# No Answer Needed: Predicting LLM Answer Accuracy from Question-Only Linear Probes

## Abstract

Do large language models (LLMs) anticipate when they will answer correctly? To study this, we extract activations *after* a question is read but *before* any tokens are generated, and train linear probes to predict whether the model's forthcoming answer will be correct. Across three open-source model families ranging from 7 to 70 billion parameters, projections on this "in-advance correctness direction" trained on generic trivia questions predict success in distribution and on diverse out-of-distribution knowledge datasets, indicating a deeper signal than dataset-specific spurious features, and outperforming black-box baselines and verbalised predicted confidence. Predictive power saturates in intermediate layers and, notably, generalisation falters on questions requiring mathematical reasoning. Moreover, for models responding "I don't know", doing so strongly correlates with the probe score, indicating that the same direction also captures confidence. By complementing previous results on truthfulness and other behaviours obtained with probes and sparse auto-encoders, our work contributes essential findings to elucidate LLM internals.

## 1 Introduction

Large language models (LLMs) internally encode information beyond what is immediately observable in their output (Burns et al., 2022; Azaria & Mitchell, 2023; Marks & Tegmark, 2023; Burger et al., 2024; Kudo et al., 2024; Goldowsky-Dill et al., 2025; Ferrando et al., 2025). Studies have demonstrated that hidden activations can reveal latent concepts related to statement truthfulness (Burns et al., 2022; Azaria & Mitchell, 2023; Marks & Tegmark, 2023; Burger et al., 2024), deception (Goldowsky-Dill et al., 2025) and hallucination (Ferrando et al., 2025).

In this work, we investigate the structure of self-correctness representations in LLMs. Specifically, we test the **Linear Representation Hypothesis** for correctness: does the residual stream activation (captured immediately after processing a query) contain a direction that linearly separates questions the model will answer correctly from those it will not? To do this, we employ a simple difference-of-means linear probe in order to verify whether the correctness signal is accessible as a linear feature of the representation space (Park et al., 2024), distinct from complex non-linear correlations (which may be more complex than necessary to find such a signal in a production setup).

Empirically, our approach identifies the activation-space vector linking the average residual stream activations for correctly answered questions to those for incorrectly answered ones (similar to Burger et al., 2024's method for statement truthfulness). We test our approach on open-source LLMs spanning three families and ranging from 7 to 70 billion parameters, and we find:

- **Linear Separability:** We confirm that a correctness signal is indeed linearly separable in the activation space. A simple linear probe trained on TriviaQA (Joshi et al., 2017) generalises to domain-specific knowledge datasets, outperforming non-linear baselines (XGBoost) that rely on model-independent question embeddings. This confirms that the internal activations contain unique, linearly accessible information regarding the model's own capabilities that is not present in the general semantic embeddings of the input.

- **Factual vs. Arithmetic Misalignment:** While the direction generalizes across factual domains (Trivia, Cities, People), it fails to generalize to mathematical reasoning (GSM8K).

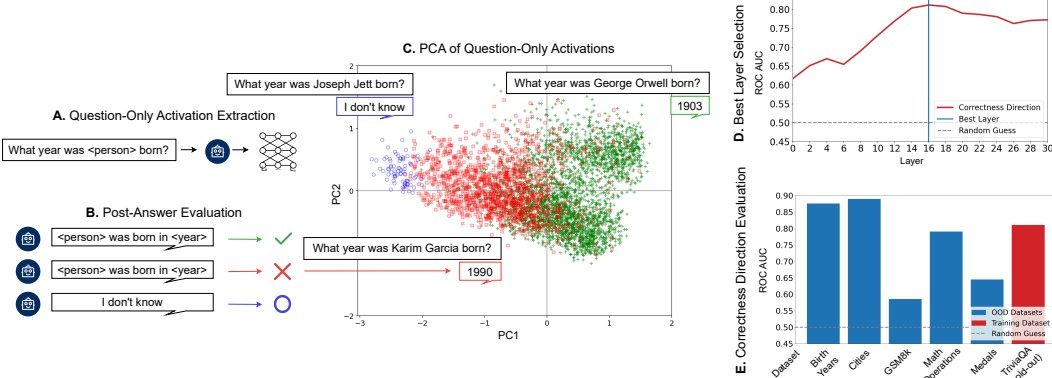

Figure 1: Proposed methodology to find the in-advance correctness direction. **(A)** Residual stream activations for all model layers are extracted at the last token of the question, prior to sampling. **(B)** Model answers are generated and evaluated against the ground truth. **(C)** The direction which mostly discriminates activations related to correct and incorrect answers is identified (the first two principal components at a specific layer are visualised). **(D)** The most discriminative layer is chosen. **(E)** The final correctness classifier is trained on the identified layer, and its out-of-distribution performance is assessed.

> This indicates that "Factual Correctness" and "Arithmetic Correctness" may be distinct, orthogonal, or structurally misaligned vectors within the model.

- **Layer-wise Emergence:** For all models, the linear separability of correctness is low in early layers and saturates at intermediate transformer layers, suggesting the model's internal assessment of the prompt crystallizes mid-computation.

- **Correlation with Abstention:** For models that answer "I don't know" without being explicitly prompted, doing so correlates with the question's position along the in-advance correctness direction, suggesting this vector also captures an implicit confidence axis.

- **Scaling Trends:** The in-advance correctness signal is strongest and most consistent for the largest model we test (Llama 3.3 70B (Touvron et al., 2023)).

Overall, our analysis advances our understanding of how LLMs encode self-assessment, providing evidence for a general "Factual Correctness" direction while highlighting the structural distinctness of reasoning capabilities. Our codebase is accessible at `https://anonymous.4open.science/r/no-answer-needed`.

## 2 RELATED WORK

By avoiding generation from the model, our approach contrasts with self-confidence estimation methods (Shorinwa et al., 2024) that consider token-level output logits (Fadeeva et al., 2024), train additional modules to predict uncertainty (Kadavath et al., 2022), measure "semantic similarity" of multiple model generations (Kuhn et al., 2023), or ask models to verbalise their uncertainty (Lin et al., 2022; Kapoor et al., 2024). Notably, there is no consensus on the performance of these methods (Kapoor et al., 2024), which were shown to be brittle to shortcuts (Heindrich et al., 2025) and to yield inconsistent results across different methods (Pawitan & Holmes, 2024). Importantly, our probe is applicable to free-form answers, while not all the above approaches are. By avoiding model generation, our approach comes close to techniques training correctness predictors using model-independent features of the input ("assessors", Hernández-Orallo et al., 2022; Zhou et al., 2024; Pacchiardi et al., 2025), but differs from those in leveraging internal representations. On the other side, in contrast to our in-advance prediction of correctness, aforementioned works using model internals mostly focused on truthfulness of complete statements (Azaria & Mitchell, 2023; Burns et al., 2022; Marks & Tegmark, 2023; Burger et al., 2024; Bao et al., 2025) or other properties such as questions answerability (Heindrich et al., 2025), deception (Yang et al., 2024; Goldowsky-Dill

et al., 2025; Parrack et al., 2025), and when in a chain of thought internals predict the answer the model will eventually produce (Kudo et al., 2024).

The closest works to ours are Kadavath et al. (2022), which tested a similar probe to older proprietary models, but did not release any code to replicate or adapt their method, and Ferrando et al. (2025), which identified the latents of pre-trained Sparse Auto-Encoders (SAEs, Bricken et al., 2023) that best distinguish questions answered correctly from those answered incorrectly in small Gemma models (Team et al., 2024). As in these works, our work has the scientific goal of obtaining a better understanding of model internals, rather than optimizing for predictive power (which can be increased by combining internal embeddings with other features, as done in Kamath et al., 2020). Detailed discussion of the above and other related works can be found in Appendix A.

## 3 METHOD

### 3.1 PROBLEM FORMULATION

Let $M$ be a LLM that, given an input prompt $x$, produces residual stream activations (after the final prompt token) $\{h^{(l)} \in \mathbb{R}^d\}$ at each layer $l = 1, \ldots, L$. For that prompt, $M$ can be used to produce (by autoregressive sampling) an output answer $y$. We define the correctness function $\text{Correct}(x, y)$ as a binary indicator of whether the answer is correct. Our objective is to learn a classifier $f_w$ (where $w$ indicates the classifier weights) that predicts the correctness label[1] from an intermediate activation $h^{(l)}(x)$, i.e.,

$$f_w\big(h^{(l)}(x)\big) \approx \mathbf{1}\{\text{Correct}(x, M(x))\}.$$

A key methodological decision in this work is the use of a simple linear classifier, specifically the difference-of-means direction, rather than more expressive non-linear probes (e.g., MLPs). We choose this to explicitly test if the correctness signal is *linearly separable*, rather than to maximize accuracy. If a simple linear direction $w$ can successfully distinguish correct from incorrect inputs across diverse datasets, it suggests that "correctness" (or confidence) is represented as a coherent feature direction in the high-dimensional activation space, supporting the Linear Representation Hypothesis (Park et al., 2024). If complex non-linear classifiers were required, it would imply the information is encoded in a more entangled manner.

### 3.2 LEARNING A LATENT CORRECTNESS DIRECTION

Taking inspiration from Burger et al. (2024), we train a simple linear probe on fixed neural activations from a single layer obtained at the final prompt token. In particular, we partition the activations into two groups according to correctness. We summarize each class by the average activation vector over all examples in that class—one centroid for the incorrect outputs, $\mu_{\text{false}}$, and one for the correct outputs, $\mu_{\text{true}}$. Then, we define the correctness direction as the difference of the centroids: $w = \mu_{\text{true}} - \mu_{\text{false}}$.

For a given activation vector $h$, we then compute its correctness score by subtracting the mean of the centroids $\mu = \frac{1}{2}\left(\mu_{\text{false}} + \mu_{\text{true}}\right)$ and projecting it on the normalized direction:

$$\text{score}(h) = \frac{(h - \mu)^\top w}{\|w\|}.$$

This score quantifies the alignment between the activation and the vector associated with correctness. Importantly, we do not apply a sigmoid to transform this score into a probability, nor use a threshold to assign class labels, although doing so is possible and straightforward. Instead, we assess the discriminative power of this direction by computing the Area Under the Receiver Operating Characteristic curve (AUROC), which is invariant under monotonic transformations of the scores and independent of any particular threshold since it measures performance across all possible cut-offs.

---

[1]If LLM answer generation is done with non-zero temperature, the correctness label inherently possesses non-zero aleatoric uncertainty (randomness). Thus, the classifier cannot perfectly predict the label, but it can approximate the random correctness label as accurately as possible, thereby reducing the epistemic uncertainty. In our experiments (Section 4), we set temperature to zero, so this consideration does not apply.

Therefore, our method does not produce a probabilistic classifier by default. It simply identifies a linear axis $w$ in activation space that optimally separates correct from incorrect outputs, and uses the projections onto this axis to evaluate their separability.

# 4 EXPERIMENTS AND RESULTS

## 4.1 SETUP

For each dataset–model pair, we collect a dataset of activations and correctness by prompting the model on every question $x$, extracting activations $h^{(l)}(x)$ (for layer $l$) at the question's final token, sampling the answer $y$ (temperature 0), and recording its correctness (0 or 1) against the gold answer. We then learn the direction as described in Section 3.2. To conduct the experiments, we used GPU resources on RunPod, employing approximately 60 hours of NVIDIA A100 time for evaluating the larger models, and around 100 hours of NVIDIA A40 time for the smaller models. We notice how most of the computational effort was spent in collecting the models' activations: our probe's training is a one-shot learning of a $d$-dimensional parameter vector on 10 k cached activations, and it takes less than three minutes on CPU; applying the probe involves a linear project, which is light-weight relatively to generation from a model.

### 4.1.1 DATASETS

We choose datasets where the performance of the considered models is in the mid range, so that enough samples are available for each of the two classes to accurately estimate the mean. Moreover, we avoid multiple-choice formats to prevent chance-correct answers from biasing our results. Instead, every dataset uses open-ended questions. Although some answer sets (e.g., countries or years) are bounded, they're broad enough that the impact of random guessing is effectively negligible.

Therefore, we select two publicly available datasets and generate a few synthetic datasets (see Table 1). In particular, we derive our largest and most diverse dataset from a subset of TriviaQA (Joshi et al., 2017), which encompasses trivia-style questions covering a wide range of topics. To complement this, we construct three datasets from public tables to evaluate the in-advance correctness directions in specific factual-knowledge domains[2].The first [3] asks which country a city belongs to; the second [4] requests to provide a notable person's birth year; and the third[5] queries which country won the gold medal in a specified sport at a particular edition of the Olympic Games. In addition, we construct our own dataset of arithmetic problems and employ GSM8K (Cobbe et al., 2021), a benchmark for mathematical reasoning.

### 4.1.2 LANGUAGE MODELS

We conduct our experiments on open state-of-the-art large language models (Table 2) varying in both training regimen and scale. We use three-shot prompting to mitigate answer formatting errors (exact prompts[6] in Appendix B.8). The performance for each model on each dataset can be found in Table 5 in Appendix B.1. Although we use a reasoning-trained model (Deepseek R1 Distill Qwen 32B), we do not employ reasoning-specific prompting and treat that identically to the other models.

### 4.1.3 BASELINES

To establish a point of reference to evaluate our approach, we consider two baseline approaches.

**Verbalized confidence.** We prompt each model to output a confidence score (0–100%) indicating its likelihood of answering each question correctly. The exact prompt can be found in Appendix B.8.

---

[2]This is because we could not find existing datasets that simultaneously offered (i) free-form answers, (ii) a narrow topical scope that lets us measure cross-domain transfer, and (iii) fully automatic grading. Our Cities, Notable People and Medals datasets satisfy all three. Although small (6 k–16 k samples) they stress the probe with questions that differ markedly from TriviaQA's trivia style.

[3]Generated from the Geonames dataset from OpenDataSoft, licensed under CC BY 4.0.

[4]Generated from the A Brief History of Human Time dataset from SciencesPo, licensed under CC-BY-SA.

[5]Generated from the Olympic History dataset on Kaggle, licensed under CC0 1.0.

[6]Exploratory investigation found the specific few-shot examples to not significantly affect performance.

Table 1: Details of the datasets employed in our work.

| Dataset | N. samples | Source | Example |
|---|---|---|---|
| TriviaQA (Joshi et al., 2017) | 60K | Public (subset) | What is the collective name of the four holy books of the Hindu religion? |
| Cities | 10K | Custom (public data) | In which country is the city of Hungerford located? |
| Notable People | 16K | Custom (public data) | What year was Thabo Mbeki (politician from South Africa) born? |
| Medals | 9K | Custom (public data) | Which country won gold in Gymnastics Men's Team All-Around in the 1948 Summer Olympics? |
| Math operations | 6K | Custom | What is 5 plus 2? |
| GSM8K (Cobbe et al., 2021) | 8K | Public | Natalia sold clips to 48 of her friends in April, and then she sold half as many clips in May. How many clips did Natalia sell altogether in April and May? |

Table 2: Large Language Models used in our work, number of transformer layers and layer achieving the best in-distribution AUC for the direction learned on TriviaQA (Section 4.2). The first layer is 0.

| Model | N. layers | Best layer |
|---|---|---|
| Llama 3.1 8B | 32 | 14 |
| Llama 3.3 70B Instruct | 80 | 76 |
| Qwen 2.5 7B Instruct | 28 | 22 |
| DeepSeek R1 Distill Qwen 32B | 64 | 44 |
| Mistral 7B Instruct v0.3 | 32 | 16 |
| Ministral 8B Instruct 2410 | 36 | 18 |

**Assessors.** We train LLM-specific binary classifiers using question text embeddings as model-independent inputs and the corresponding evaluated model answers as labels. These black-box *assessors* (Hernández-Orallo et al., 2022) predict an LLM's performance on unseen questions based on the question's embedded features. Following Pacchiardi et al. (2024), we use OpenAI's `openai_text-embedding-3-large` model to obtain 3,072-dimensional question embeddings, and we explore logistic regression and gradient boosted decision trees (Chen & Guestrin, 2016) to establish linear and non-linear baseline assessors, respectively.

## 4.2 IDENTIFYING THE MOST DISCRIMINATIVE LAYER

For each LLM, we first identify the layer that most effectively discriminates between questions the model answers correctly and those it answers incorrectly, with the approach in Section 3.2. We perform this evaluation on TriviaQA because it offers a diverse array of questions across multiple domains and complexity levels, which mitigates the risk of discovering an activation direction tied to features merely correlated with model success rather than the model's internal correctness prediction. The remaining datasets are kept held-out for further evaluation. Appendix B.7 contains similar experiments with all other datasets.

Thus, we dedicate a subset of 10,000 samples from TriviaQA exclusively to this step. We collect activation samples every 2 layers for small (<10B parameters) models and every 4 for larger (>10B parameters) models. On this data, for each model and layer, we perform 3-fold cross-validation and train the model described in Section 3.2. Figure 2 presents the average AUROC over folds, and Table 2 lists the resulting optimal layers. We observe that the early layers generally perform poorly and performance saturates around the midpoint, with the optimal layer typically lying between the midpoint and the final layer. This suggests the model's understanding of its own answering ability emerges progressively across layers, consistent with Ferrando et al. (2025) and Burger et al. (2024), who also found representations in the middle layers to perform better for their task.

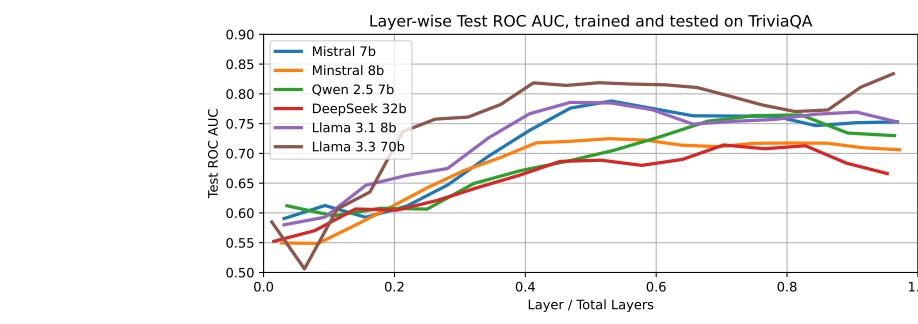

Figure 2: TriviaQA AUROC (average over 3 folds) across layers. We collect activations every 2 layers for small (<10B parameters) models and every 4 layers for large (>10B parameters) models.

### 4.3 CORRECTNESS DIRECTION GENERALIZATION

#### 4.3.1 TRAINING ON TRIVIAQA

After identifying each model's most informative layer on TriviaQA (Section 4.2), we evaluate whether the corresponding in-advance correctness direction generalises to other datasets. To do so, we split each dataset (excluding the first 10,000 elements of TriviaQA, used for layer selection, Section 4.2) into 5 folds, we train the correctness direction by iteratively considering 4 folds of TriviaQA and evaluating on the remaining one (for in-distribution performance) and on one individual fold for each of the out-of-distribution (OOD) datasets[7]. Similarly, we train the assessor baselines on TriviaQA and test them on the other datasets (the confidence baseline instead requires no training set). Table 3 reports the resulting average AUROC for each method: the correctness direction found on TriviaQA demonstrates significantly stronger generalization to all other datasets, with the exception of GSM8K, than the baseline methods (Section 4.1.3), despite being slightly outperformed by the logistic regression assessor in distribution. The direction approach improves on the best baseline by 10–22 AUROC points on Notable People, 5–18 points on Cities and 28–39 points on Math-Operations. On Medals, it remains the strongest method, but the margin contracts to 4–15 points, indicating that the gain diminishes on this harder multi-hop task. These patterns suggest that recognising one's own competence scales with question difficulty and with model size: only the 70 B model shows a sizeable advantage on Medals. In contrast, no method can skilfully predict any of the models' success on GSM8K well, confirming that the correctness signal does not transfer to arithmetic-reasoning tasks and indicating a potential limitation of current models. In Table 6, the standard deviation over the 5 folds for the learned direction is reported, showing that this is smaller than 0.035.

In Figure 5, we analyze the sample efficiency of the linear probe. Identifying the correctness direction requires very little data: robust performance is achieved with as few as 160 samples, and 2,560 samples are sufficient to match the performance obtained using the full 48,540 TriviaQA dataset. This high sample efficiency provides strong support for the Linear Representation Hypothesis: if the correctness signal were encoded in a complex, highly non-linear manifold, a simple difference-of-means probe would likely require significantly more data to generalize effectively. Notably, larger models require fewer samples to converge, suggesting that as model scale increases, the internal representation of correctness becomes more distinct and linearly accessible. Conversely, performance on GSM8K plateaus near random chance regardless of dataset size, reinforcing that the "Factual Correctness" direction is structurally misaligned with reasoning tasks, rather than simply being noisy or under-sampled.

We observe that while the Assessor baselines (trained on OpenAI embeddings) achieve slightly higher in-distribution scores, they suffer significant degradation out-of-distribution compared to our

---

[7]This is done so that the results are comparable to the ones obtained by training the correctness direction on the OOD dataset–which are too small for cross-validation to be applied–as discussed immediately below. Throughout training we always test on the held-out fold of the dataset whose AUROC we report, however, we train the probe in two alternative ways: (i) on the four remaining folds of TriviaQA (to assess cross-domain generalisation) or (ii) on the four folds of that same OOD dataset as an upper-bound. Hence, every cell of Table 3 compares methods on identical test questions, irrespective of training set.

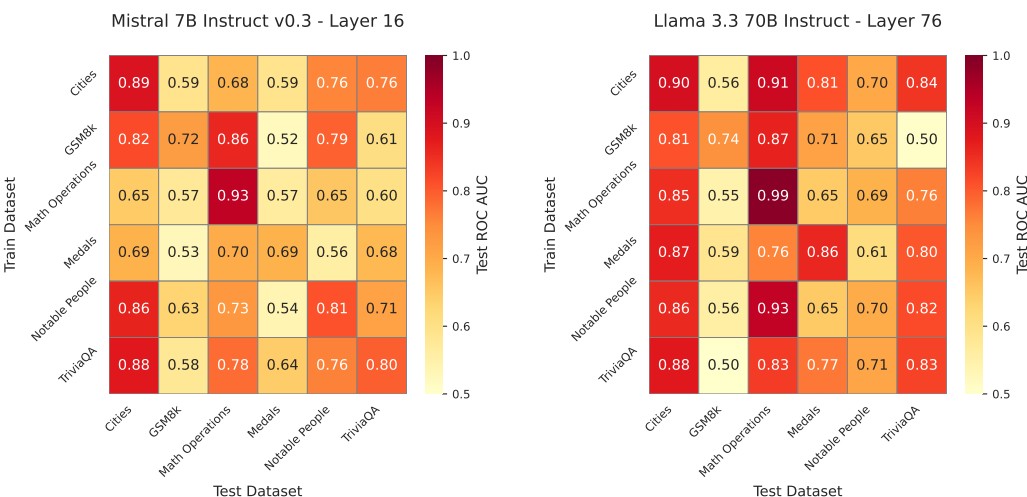

Figure 3: AUROC scores on each dataset for the direction learned on each dataset individually, for two selected models (others in Appendix B.5). Average AUROC over 5 folds is reported (Section 4.3).

linear probe. This contrast highlights the nature of the signal: the Assessor likely relies on spurious correlations in the semantic embeddings of the question text, which fail to hold across domain shifts. In contrast, our linear probe targets the model's internal state. The fact that a simple linear direction on internal activations generalizes better than a non-linear XGBoost classifier on external embeddings (see Appendix B.3) confirms that the residual stream contains a genuine, transferable signal of self-competence that is not present in the surface semantics of the question alone.

### 4.3.2 TRAINING ON OTHER DATASETS

Next, to understand if the generalization is due to training on TriviaQA or if it can be instead obtained with any other dataset, we train and test the correctness direction on all dataset combinations, keeping the same folds as described above. We report results for Mistral 7B Instruct and Llama 3.3 70B in Figure 3 (other models in Appendix B.5). In some cases, the direction learned on the smaller specific datasets (such as Cities and Notable People) transfers well to others, but this does not always happen; at the same time, all datasets lead to decent in-distribution performance, with Medals being often the lowest one (even considering the other models in Appendix B.5). In Appendix B.6, we further report the cosine similarity between the directions learned on the different datasets, which shows how the directions learned on the small datasets are mostly orthogonal, except for a few cases (Cities and Notable People), which are also more aligned with the direction learned on TriviaQA. These observations suggest that, for certain datasets, the learned direction captures dataset-specific cues correlated with correctness rather than correctness itself[8]. Over all models, moreover, the direction learned from TriviaQA shows the strongest generalisation (except for GSM8K and Medals, as already discussed above), likely because its diverse nature makes it less likely to contain exploitable dataset-specific patterns. Further, comparing the models, we find that Llama 3.3 70B generalizes well across the largest number of train–test dataset pairs, suggesting that this larger model has a more consistent correctness direction.

### 4.4 QUALITATIVE INVESTIGATION

Alongside our main experiments, we observe several behaviours and patterns that are not easily quantifiable but nonetheless offer a valuable insight into the quality of the direction we find. In particular, Section 4.5 shows how some models, despite being urged to give an answer in a specific format, produced some form of abstention; these answers are located on the negative extreme of the

---

[8]For context, QA accuracies on GSM8K range from 10.7% to 44.3% and on Cities from 45.8% to 80.3% (Table 5), so all datasets have substantial mass in both classes, an 'always correct' baseline would have AUROC 0.5 by construction.

Table 3: AUROC for each dataset, model and method. All directions are trained on the TriviaQA dataset on the optimal layer found in Section 4.2. Average AUROC over 5 folds is reported (Section 4.3, variance reported in Table 6). For the assessors, we only report the best performing one (logistic regression; all results in Appendix B.3). As the probes are trained on finite samples, weak or noisy signal might result in AUROC slightly below 0.5.

| Model | Method | Test dataset | | | | | |
|---|---|---|---|---|---|---|---|
| | | TriviaQA | N. people | Cities | Math ops. | Medals | GSM8K |
| Llama 3.1 8B | Assessor | **0.852** | 0.630 | 0.663 | 0.528 | 0.623 | **0.558** |
| | Verb. conf. | 0.502 | 0.499 | 0.500 | 0.623 | 0.500 | 0.540 |
| | Direction | 0.804 | **0.722** | **0.732** | **0.858** | **0.680** | 0.534 |
| Llama 3.3 70B Instruct | Assessor | 0.759 | 0.583 | 0.672 | 0.449 | 0.568 | 0.573 |
| | Verb. conf. | 0.580 | 0.594 | 0.694 | **0.913** | 0.665 | **0.598** |
| | Direction | **0.826** | **0.708** | **0.880** | 0.835 | **0.770** | 0.499 |
| Qwen 2.5 7B Instruct | Assessor | **0.807** | 0.723 | 0.708 | 0.400 | **0.622** | 0.584 |
| | Verb. conf. | 0.643 | 0.637 | 0.758 | 0.517 | 0.531 | 0.513 |
| | Direction | 0.758 | **0.800** | **0.842** | **0.837** | 0.586 | **0.601** |
| DeepSeek R1 Distill Qwen 32B | Assessor | **0.790** | 0.709 | 0.663 | 0.337 | 0.601 | **0.576** |
| | Verb. conf. | 0.619 | 0.605 | 0.577 | 0.499 | 0.563 | 0.503 |
| | Direction | 0.735 | **0.825** | **0.879** | **0.847** | **0.638** | 0.552 |
| Mistral 7B Instruct v0.3 | Assessor | **0.846** | 0.673 | 0.710 | 0.493 | 0.638 | 0.559 |
| | Verb. conf. | 0.570 | 0.625 | 0.705 | 0.617 | 0.558 | 0.525 |
| | Direction | 0.796 | **0.760** | **0.880** | **0.782** | **0.645** | **0.579** |
| Ministral 8B Instruct 2410 | Assessor | **0.789** | 0.623 | 0.682 | 0.454 | 0.626 | **0.598** |
| | Verb. conf. | 0.515 | 0.500 | 0.554 | 0.500 | 0.502 | 0.577 |
| | Direction | 0.734 | **0.680** | **0.840** | **0.844** | **0.670** | 0.578 |

correctness direction, and suggests that our correctness-prediction direction also captures abstention. Instead, Section 4.6 contains a manual investigation of correct and incorrect answers with the highest positive and negative values of correctness scores, showing patterns which intuitively align with confidence scores (e.g., wrong answers with high scores being near-misses where the model fails by one or a few years).

### 4.5 LOCATION OF "I DON'T KNOW" RESPONSES

Some of the models we tested, despite being urged to give a specific answer format by our prompt, produced answers of the form "I don't know" (IDK) or similar to some questions. When training the correctness direction, these were considered as incorrect answers. By visualizing (Figure 4) the distribution of activation projections on the correctness direction at the optimal layer found in Section 4.2, we see that the questions where the model answers IDK are consistently located more at the negative extreme of the correctness direction than the questions where the model attempts an answer but fails. This behaviour demonstrates that the overall internal state, causally upstream of the model outputting "I don't know" or attempt an answer, is strongly captured in the direction that we find. This aligns with Ferrando et al. (2025)'s finding of "knowledge-awareness" directions causally affecting answer refusal. Thus, our "correctness-prediction" direction could also be interpreted as a confidence direction: the model will only say that it doesn't know if its confidence on whether it can answer the question is very low.

### 4.6 MANUAL INVESTIGATION OF EXTREME VALUES

Finally, we report in Table 4 the correct and incorrect answer with the highest positive and negative values of correctness scores for Mistral 7B Instruct. The patterns we observe are intuitive: among the incorrect answers with low confidence scores, we find IDK responses, which is consistent with the behaviour discussed in Section 4.5. For wrong answers with high scores, we often see questions for which the model fails by one or a few years, and the correct answers with the highest confidence

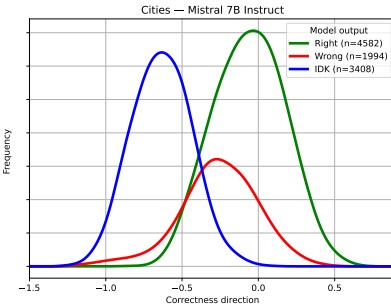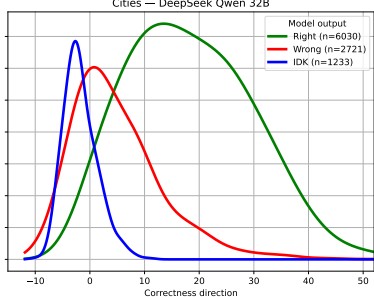

Figure 4: Distribution of values of activation projections on the correctness direction from TriviaQA, grouped by produced answer (right, wrong,"I don't know"), for a selection of models and datasets.

involve very well-known individuals, which aligns with the interpretation that we are finding a confidence direction.

Table 4: Questions of the Notable People dataset with the most extreme values on the correctness direction trained on TriviaQA for Mistral 7B Instruct.

|  |  | Person | Answer (Correct) |
|---|---|---|---|
| Wrong answers | Low scores | Victoria (Royal Family from Germany) | IDK (1840) |
|  |  | Yokozuna (wrestler from US) | 1972 (1966) |
|  | High scores | Kazimir Malevich (painter from Russia) | 1961 (1962) |
|  |  | A. A. Milne (writer from United Kingdom) | 1892 (1882) |
| Right answers | Low scores | Jim Carter (actor from United Kingdom) | 1948 (1948) |
|  |  | David Keith (film-maker from US) | 1954 (1954) |
|  | High scores | Charles Darwin (biologist from United Kingdom) | 1809 (1809) |
|  |  | Albert Einstein (physicist from Germany) | 1879 (1879) |

## 5 CONCLUSION

We have provided evidence for the Linear Representation Hypothesis regarding LLM self-correctness. By analyzing the residual stream before token generation, we identified a "correctness direction" that is linearly separable and generalizes across diverse factual tasks. Our comparison with external-embedding baselines confirms that this signal is intrinsic to the model's internal state. However, the failure to generalize to mathematical reasoning suggests that factual retrieval and arithmetic reasoning may rely on distinct internal verification mechanisms. These findings advance our scientific understanding of how LLMs encode their own capabilities and limitations.

Specifically: (1) we provide evidence that LLMs embed a latent correctness signal mid-computation; (2) we show that a simple linear probe can extract this signal (providing evidence for linear separability), yielding generalisation across knowledge datasets; (3) we highlight the limits of this approach, suggesting that deeper reasoning and arithmetic capabilities are not as easily captured in activations; (4) we find a stronger signal for the largest model we test (Llama 3.3 70B), suggesting that larger models may better predict their correctness, and (5) we demonstrate this direction aligns with abstention behaviour in models that say "I don't know," supporting its interpretation as a latent confidence axis.

This work contributes to mechanistic interpretability by identifying a meaningful confidence direction within LLM activations, corroborating recent works with sparse auto-encoders (Ferrando et al., 2025). It also complements studies of truthfulness and hallucination, suggesting that models encode internal notions of confidence—even before answer generation—and truthfulness that are both general and accessible.

Our findings have relevance for both AI safety and practical deployment. As LLMs are increasingly used in high-stakes settings, low-cost internal signals of impending failure offer a path toward safer, more robust systems. The correctness direction, combined with an ensemble of other black- and white-box methods, could inform early stopping, fallback mechanisms, or human-in-the-loop protocols, particularly where generating unreliable outputs is costly or dangerous.

## 6 LIMITATIONS

**Correctness is represented as a binary label from a single sample.** This ignores stochasticity of answer generation and the existence of questions with ambiguous or nuanced answers that cannot be captured by a single true/false label. Future work might involve generating multiple samples or assigning real-valued correctness scores to obtain a more robust estimate of expected correctness; a natural extension for reasoning specifically is to probe along the reasoning trajectory and to compare linear probes against stronger baselines on reasoning-heavy datasets.

**Linear probes may underestimate predictive power.** We used linear representations as a model can more conceivably access them during answer generation, but higher-capacity non-linear classifiers may yield greater predictive power. Further, the probes are targeted towards a pre-generation factual correctness signal, which is not a memorisation-free notion of correctness, and should be carefully evaluated in future work.

**Model diversity and scale.** We evaluate on six open-source models from three families, up to 70B parameters. While this spans a wide range, results may not fully generalize to proprietary models, alternative architectures (e.g., mixture-of-experts), or future frontier systems. Only a single large model (70B) was used due to compute limitations.

**Layer selection is performed on a single dataset.** We identify the most discriminative layer for each model using a single large general dataset (TriviaQA). To ensure the generality of our findings, we could repeat the layer choice on other large generalist datasets.

## 7 REPRODUCIBILITY STATEMENT

We provide the codebase, raw datasets, and generated datasets to fully reproduce our work at `https://anonymous.4open.science/r/no-answer-needed`. The recorded activations for all our models will be made available upon that paper's acceptance (as their size prevents us from adding them to the anonymous repository).

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

## A FURTHER RELATED WORK

### A.1 UNCERTAINTY QUANTIFICATION AND CONFIDENCE ESTIMATION IN LLMS

Traditional uncertainty quantification approaches for deep learning models include looking at the logits of a multi-class classification network (Guo et al., 2017) or training the model to embed a form of uncertainty (such as Bayesian Neural Networks, (Jospin et al., 2022) or Dropout (Gal & Ghahramani, 2016)). Some of these methods can be adapted to LLMs (Shorinwa et al., 2024), for instance by considering token-level output logits (Kadavath et al., 2022; Fadeeva et al., 2024) or training additional modules to predict uncertainty (Kadavath et al., 2022). In simple classification tasks with single-token or multiple-choice answers, these probabilities often track the LLM's confidence. However, extending these methods to open-ended answers is non-trivial, as low-level probabilities do not necessarily yield a clear answer-level confidence. To this end, methods based on "semantic similarity" (Kuhn et al., 2023) were proposed, but these are costly as they require the generation of multiple answers from the model. A more recent approach (Sam et al., 2025) asks a set of "elicitation questions" after an answer is generated and uses the responses to predict its correctness; as for semantic similarity approaches, this requires repeated model queries.

Alternatively, LLMs can be asked to explicitly verbalize their uncertainty, with or (rarely) without reference to a specific answer. For instance, (Lin et al., 2022) and Kapoor et al. (2024) finetuned LLMs to numerically report their belief in the correctness of an answer, while Kadavath et al. (2022) showed that older Anthropic models can verbally quantify their uncertainty about answers to multiple-choice questions and finetuned the models to predict the confidence of succeeding on a question without reference to a specific answer, which performed satisfactorily but struggled with novel tasks. Instead, Chaudhry et al. (2024) finetuned LLMs to emit linguistic expressions of uncertainty (e.g., "maybe"). Finally, arguing that absolute verbalised confidence estimation is poor, Shrivastava et al. (2025) prompted LLMs to estimate whether they are more confident in their answers to one question relative to another one, then aggregate many of these pairwise comparisons into confidence scores using a ranking procedure, finding small gains in discriminative power.

Notably, there is no consensus on the performance of these methods (Kapoor et al., 2024), which were shown to be brittle to shortcuts (Heindrich et al., 2025) and to yield inconsistent results across different methods (Pawitan & Holmes, 2024). In contrast to the methods above, we directly leverage previously trained LLM internals *before an answer is generated*, making our method applicable to free-form answers and avoiding generation from the model. Further, while sampling-based confidence estimators would typically need 10-20 candidate answers per query; our probe caches activations in a single forward pass, then applies a cheap linear projection. Probe training is a one-shot learning of a $d$-dimensional parameter vector on 10 k cached activations (<3 min on CPU); applying the probe involves a linear project, which is drastically lighter-weight in deployment. While the method assumes white-box access, it complements black-box sampling: practitioners can choose probes when speed or token budget is paramount and revert to sampling when internals are unavailable.

### A.2 ANTICIPATING LLM PERFORMANCE

Our approach aims to anticipate LLM's performance based on its internals before an answer is generated. Some works (Hernández-Orallo et al., 2022; Zhou et al., 2022; Schellaert et al., 2024; Pacchiardi et al., 2024; 2025) attempted to predict LLM performance by training independent score predictors ("assessors") based on features of the input question obtained independently of the considered LLM. This is motivated by the idea of "Predictable AI" (Zhou et al., 2024), which argues that predicting the inputs on which an AI system will behave as expected is a necessary component of safety. Our work can be seen as belonging to this research strand, with the key distinction of leveraging model internals, which provide more information than model-independent features.

A few works leveraged internals to predict models' ability to answer a question correctly, but no work has investigated directly training linear probes only relying on internals. First, Kamath et al. (2020) combined embeddings generated by a model prompted with a question with hand-crafted feature and the (anticipative) confidence scores of the model, and trained non-linear models (such as XGBoost) to predict correctness in advance of generating answers. In contrast, our approach uses only linear probes on model internals to determine whether activations from correctly and incorrectly answered questions are linearly separable. More recently, Ferrando et al. (2025) contains two experiments using

the latent representations of SAEs (pre-trained to reconstruct model representations in unsupervised manner) on base Gemma2 2B and 9B and LLaMA 3.1 8B: in the first, they identified general "knowledge-awareness" directions that predominantly activate on known entities but not on unknown ones (and vice versa) and showed that steering the chat-fine-tuned version of the model using these directions induces hallucination or refusal. This parallels our finding that the activations lie at an extreme of the direction we identify when the model utters "I don't know" (Section 4.5). In the second experiment, closer to our setup, they posed questions to the chat model, excluded cases where the model refuses to respond, and identified the SAE latent that has the highest difference in values between when the model produces correct and incorrect answers ("uncertainty direction"), and found good predictive power. Notably, this analysis was confined to Gemma2 models; by contrast, we directly train simple linear directions across a broader range of models, scaling up to 70 billion parameters. Nevertheless, Ferrando et al.'s approach and ours produce consistent evidence, thereby reaffirming one another.

Several contemporaneous works also use LLM internals to anticipate hallucination or non-factuality at question time (Snyder et al., 2024; Gottesman & Geva, 2024; Wang et al., 2024; Ji et al., 2024; Slobodkin et al., 2023). Snyder et al. (2024) and Ji et al. (2024) focus on hallucination risk and often use richer information about the generation process, such as the first generated token or short trajectories. Slobodkin et al. (2023) study (un)answerability in reading comprehension with an explicit context passage. Gottesman & Geva (2024) shift the focus to the level of entities, and estimate whether a model "knows" facts about specific entities such as historical figures. Wang et al. (2024) are closest in spirit to our work, and train more expressive classifiers over internal states to predict non-factuality and to transfer signals across models. In contrast, we deliberately adopt a stricter and simpler setting. We work with a single one-dimensional difference-of-means direction in the residual stream of one layer, trained on question-only activations before any token is generated, and we study per-question self-correctness for free-form factual QA across six open-source models from three families. This minimal probe already predicts correctness across datasets and models, which strengthens the claim that a linearly accessible correctness or confidence signal exists in mid–late layers.

### A.3 Probing other properties with Model Internals

Closely related to our work is the growing literature on using hidden activations to detect properties of the model's upcoming or generated outputs. Several papers (Burns et al., 2022; Azaria & Mitchell, 2023; Marks & Tegmark, 2023; Burger et al., 2024; Bao et al., 2025) showed that linear or shallow probes on internal representations can detect whether a full statement (or question+answer) processed by the model is true or false. Analogously to our findings, Bao et al. (2025) found that more capable models have stronger representations, and that probes trained on atomic statements generalise to more complex sentences. This is closely related to Concept Activation Vectors (Kim et al., 2018) in explainable AI, which describe a model's internal representations in terms of its sensitivity to user-defined examples for a concept. Our approach chiefly differs from those mentioned above by relying on the activations obtained *in advance* of the model generating an answer to a question. Experiments in this setup where conducted (among other things) in Kadavath et al. (2022) for proprietary models. Our work corroborates their promising results with evidence from newer open-source LLMs.

Other works extracted other information from internals. For instance, Heindrich et al. (2025) predicted question answerability. Our work focuses on correctness, which encapsulates when a question is possibly answerable but the model is incorrect. Kudo et al. (2024) studied internals across models' chain of thought and analyse when they start predicting the answer the model eventually produces; in contrast, we study whether the internals predict *correctness* of the answer. Goldowsky-Dill et al. (2025) studied the detection of deception (a model deliberately misleading its interlocutor); correctness is broader, and less dependent on the fragile role-play setting required to elicit such deceptive behaviour. It is worth noting that Parrack et al. (2025) found white box probes to be slightly better than black-box detection approaches. Relatedly, Yang et al. (2024) analyse how LLMs internally separate truthful vs. lying scenarios (with the model instructed to lie) at different layers, using dimensionality reduction and intervention experiments ("patching" activations from a lying scenario into a truthful one). Instead, (Nguyen et al., 2025) show that linear probes can separate tasks encountered during evaluations and deployment.

Finally, Lee et al. (2025) used activation steering to condition models to refuse harmful prompts; our work focuses on correctness rather than harmfulness, which requires understanding one's own capabilities. In certain cases, we expect correctness to be a prerequisite to harmfulness, as incorrectly answering a banal question may be harmful but correctly answering it may be completely safe. Beaglehole et al. (2025) conducted a similar steering study, detecting semantic concepts using non-linear feature learning and aggregating features across layers. In contrast, we show that correctness, a particularly important semantic concept, can be captured using linear features in individual layers, indicating that the concept is strongly present and can be easily accessed by the model.

# B    ADDITIONAL QUANTITATIVE RESULTS

## B.1    LLM PERFORMANCE ON DATASETS

Table 5 reports performance of all models on each dataset with the prompts we used (Appendix B.8).

Table 5: Model performance across tasks (%).

| Model | TriviaQA | N. people | Cities | Math ops. | Medals | GSM8K |
|---|---|---|---|---|---|---|
| Llama 3.1 8B | 85.6 | 93.4 | 67.4 | 77.5 | 46.0 | 13.3 |
| Mistral 7B Instruct v0.3 | 83.6 | 84.7 | 45.8 | 73.9 | 42.8 | 10.7 |
| Llama 3.3 70B Instruct | 93.4 | 97.6 | 80.3 | 82.2 | 83.5 | 36.3 |
| Qwen 2.5 7B Instruct | 73.8 | 42.8 | 62.9 | 80.8 | 29.8 | 36.0 |
| Ministral 8B Instruct 2410 | 79.9 | 67.4 | 73.0 | 74.2 | 40.5 | 14.8 |
| DeepSeek R1 Distill Qwen 32B | 59.9 | 50.8 | 60.3 | 82.0 | 33.0 | 44.3 |

## B.2    EXTENDED INFORMATION ON THE DIRECTION APPROACH

Extended experimental results on the direction approach are provided in this section, offering further context that was not included in the main text. Table 6 extends the results of the direction approach presented in Table 3 and Table 7 provides the accuracy on all test datasets of a potential classifier derived from the correctness direction.

Table 8 shows the AUROC values on the model optimal layer (from Section 4.2) and the AUROC values on the layer optimized per dataset for all models and datasets. This comparison demonstrates that the single model-optimal layer is already highly effective, achieving performance very close to the dataset-specific optimal layer.

Table 6: Mean and standard deviation AUROC of the 5 folds for the direction approach (Section 3.2) for each dataset and model. All directions are trained on the TriviaQA dataset on the optimal layer found in Section 4.2.

| Model | Test dataset | | | | | |
| | TriviaQA | N. people | Cities | Math ops. | Medals | GSM8K |
|---|---|---|---|---|---|---|
| Llama 3.1 8B | 0.804 ± 0.006 | 0.722 ± 0.010 | 0.732 ± 0.018 | 0.858 ± 0.027 | 0.680 ± 0.007 | 0.534 ± 0.022 |
| Llama 3.3 70B | 0.826 ± 0.006 | 0.708 ± 0.018 | 0.880 ± 0.014 | 0.835 ± 0.031 | 0.770 ± 0.022 | 0.499 ± 0.015 |
| Qwen 2.5 7B | 0.758 ± 0.006 | 0.800 ± 0.013 | 0.842 ± 0.008 | 0.837 ± 0.032 | 0.586 ± 0.014 | 0.601 ± 0.015 |
| Deepseek R1 32B | 0.735 ± 0.005 | 0.825 ± 0.008 | 0.879 ± 0.007 | 0.847 ± 0.035 | 0.638 ± 0.020 | 0.552 ± 0.012 |
| Mistral 7B | 0.796 ± 0.009 | 0.760 ± 0.016 | 0.880 ± 0.008 | 0.782 ± 0.033 | 0.645 ± 0.005 | 0.579 ± 0.016 |
| Ministral 8B | 0.734 ± 0.004 | 0.680 ± 0.007 | 0.840 ± 0.021 | 0.844 ± 0.020 | 0.670 ± 0.015 | 0.578 ± 0.013 |

## B.3    ASSESSORS PERFORMANCE

For the gradient boosted decision tree assessors, we used XGBoost (Chen & Guestrin, 2016). The number of trees were chosen individually for each model by performing 5-fold cross validation on the same training subset of TriviaQA as in Section 4.3. The rest of XGBoost's hyperparameters were left as default.

Table 7: Accuracy of a classifier based on the correctness direction at the optimal layer from Section 4.2. For each test dataset, we display results for the direction trained on TriviaQA and for the direction trained on the dataset itself. The threshold for the classifier is chosen using only training data.

| Model | Test dataset | | | | | | | | | | | |
| | TriviaQA | | N. people | | Cities | | Math ops. | | Medals | | GSM8K | |
| Trained with | TriviaQA | Itself | TriviaQA | Itself | TriviaQA | Itself | TriviaQA | Itself | TriviaQA | Itself | TriviaQA | Itself |
|---|---|---|---|---|---|---|---|---|---|---|---|---|
| Llama 3.1 8B | 0.728 | 0.728 | 0.522 | 0.688 | 0.551 | 0.714 | 0.755 | 0.874 | 0.612 | 0.682 | 0.500 | 0.670 |
| Llama 3.3 70B | 0.750 | 0.750 | 0.621 | 0.651 | 0.797 | 0.808 | 0.520 | 0.943 | 0.553 | 0.782 | 0.500 | 0.671 |
| Qwen 2.5 7B | 0.688 | 0.688 | 0.639 | 0.764 | 0.515 | 0.762 | 0.792 | 0.938 | 0.512 | 0.640 | 0.501 | 0.654 |
| Deepseek R1 32B | 0.671 | 0.671 | 0.588 | 0.793 | 0.612 | 0.802 | 0.848 | 0.961 | 0.519 | 0.636 | 0.500 | 0.660 |
| Mistral 7B | 0.721 | 0.721 | 0.523 | 0.724 | 0.771 | 0.811 | 0.724 | 0.858 | 0.545 | 0.661 | 0.503 | 0.659 |
| Ministral 8B | 0.670 | 0.670 | 0.514 | 0.647 | 0.500 | 0.774 | 0.764 | 0.830 | 0.532 | 0.656 | 0.502 | 0.671 |

Table 8: AUROC values on the model optimal layer from Section 4.2 and on the layer optimized per dataset for all models and datasets.

| Model | | Test dataset | | | | | | | | | |
| | | TriviaQA | | Cities | | Math ops. | | Medals | | GSM8K | |
| Opt. layer over | | Model | Dataset | Model | Dataset | Model | Dataset | Model | Dataset | Model | Dataset |
|---|---|---|---|---|---|---|---|---|---|---|---|
| Llama 3.1 8B | Layer | 14 | 16 | 14 | 30 | 14 | 12 | 14 | 28 | 14 | 12 |
| | AUROC | 0.802 | 0.803 | 0.732 | 0.783 | 0.860 | 0.901 | 0.677 | 0.712 | 0.537 | 0.548 |
| Llama 3.3 70B | Layer | 76 | 76 | 76 | 72 | 76 | 32 | 76 | 40 | 76 | 32 |
| | AUROC | 0.821 | 0.821 | 0.881 | 0.890 | 0.835 | 0.979 | 0.772 | 0.794 | 0.499 | 0.625 |
| Qwen 2.5 7B | Layer | 22 | 22 | 22 | 20 | 22 | 20 | 22 | 20 | 22 | 20 |
| | AUROC | 0.760 | 0.760 | 0.841 | 0.854 | 0.833 | 0.875 | 0.587 | 0.588 | 0.602 | 0.678 |
| Deepseek R1 32B | Layer | 44 | 44 | 44 | 48 | 44 | 52 | 44 | 48 | 44 | 12 |
| | AUROC | 0.734 | 0.734 | 0.880 | 0.890 | 0.848 | 0.963 | 0.640 | 0.665 | 0.554 | 0.642 |
| Mistral 7B | Layer | 16 | 16 | 16 | 30 | 16 | 22 | 16 | 20 | 16 | 18 |
| | AUROC | 0.798 | 0.798 | 0.879 | 0.914 | 0.801 | 0.915 | 0.647 | 0.699 | 0.579 | 0.592 |
| Ministral 8B | Layer | 18 | 18 | 18 | 16 | 18 | 16 | 18 | 16 | 18 | 2 |
| | AUROC | 0.738 | 0.738 | 0.840 | 0.846 | 0.843 | 0.859 | 0.666 | 0.669 | 0.574 | 0.637 |

Table 9: AUROC for logistic regression and Gradient Boosted Decision Tree (XGBoost) assessors.

| Model | Assessor | Test dataset | | | | | |
| | | TriviaQA | N. people | Cities | Math ops. | Medals | GSM8K |
|---|---|---|---|---|---|---|---|
| Llama 3.1 8B | Log. regression | 0.852 | 0.630 | 0.663 | 0.528 | 0.623 | 0.558 |
| | XGBoost (133 trees) | 0.896 | 0.560 | 0.639 | 0.453 | 0.554 | 0.532 |
| Llama 3.3 70B Instruct | Log. regression | 0.759 | 0.583 | 0.672 | 0.449 | 0.568 | 0.573 |
| | XGBoost (150 trees) | 0.853 | 0.516 | 0.608 | 0.398 | 0.501 | 0.543 |
| Qwen 2.5 7B Instruct | Log. regression | 0.807 | 0.723 | 0.708 | 0.400 | 0.622 | 0.584 |
| | XGBoost (47 trees) | 0.847 | 0.619 | 0.624 | 0.506 | 0.580 | 0.546 |
| DeepSeek R1 Distill Qwen 32B | Log. regression | 0.790 | 0.709 | 0.663 | 0.337 | 0.601 | 0.576 |
| | XGBoost (51 trees) | 0.834 | 0.608 | 0.609 | 0.458 | 0.547 | 0.541 |
| Mistral 7B Instruct v0.3 | Log. regression | 0.846 | 0.673 | 0.710 | 0.493 | 0.638 | 0.559 |
| | XGBoost (130 trees) | 0.898 | 0.558 | 0.672 | 0.380 | 0.590 | 0.543 |
| Ministral 8B Instruct 2410 | Log. regression | 0.789 | 0.623 | 0.682 | 0.454 | 0.626 | 0.598 |
| | XGBoost (65 trees) | 0.846 | 0.545 | 0.611 | 0.498 | 0.551 | 0.556 |

## B.4 HOW MUCH TRAINING DATA DO WE NEED TO LEARN THE CORRECTNESS DIRECTION?

Figure 5 shows performance for the correctness direction trained on TriviaQA for an increasing number of training samples. Interestingly, Mathematical Operations has the highest data complexity, likely due to the fact that arithmetic errors are heteregenous and need a large amount of averaging out to cancel the variance of the activations.

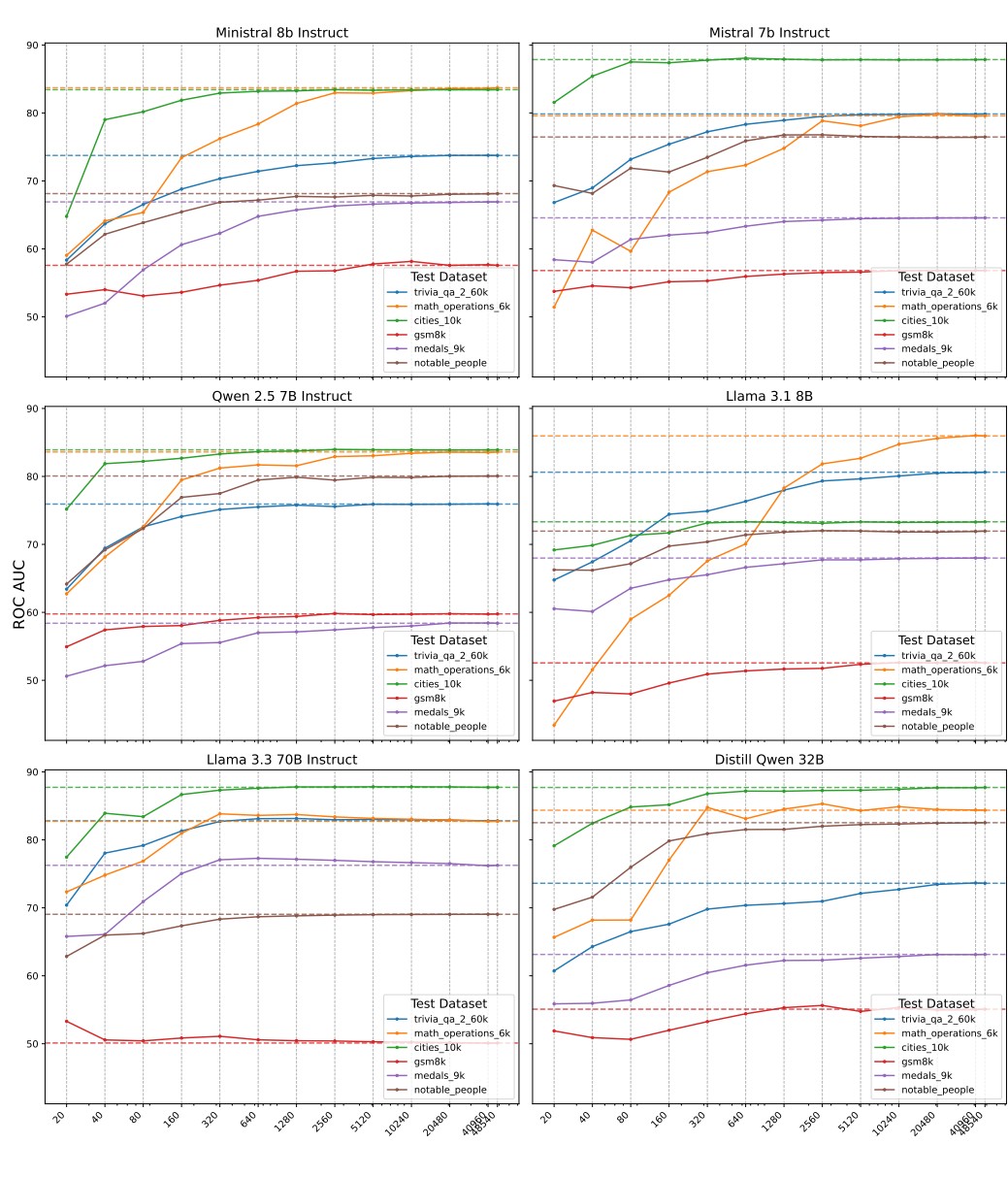

Figure 5: AUROC scores for each model and test dataset for different number of training samples from TriviaQA, for our correctness direction approach. To reduce variance, 10 experiments were performed for each number of training samples and the average AUROC is reported. Notice that the x scale is logarithmic.

## B.5 HEATMAPS

Figure 6 and Figure 7 complement Figure 3 from the main text and reports AUROC mean and standard deviation scores for each combination of model, train dataset and test dataset.

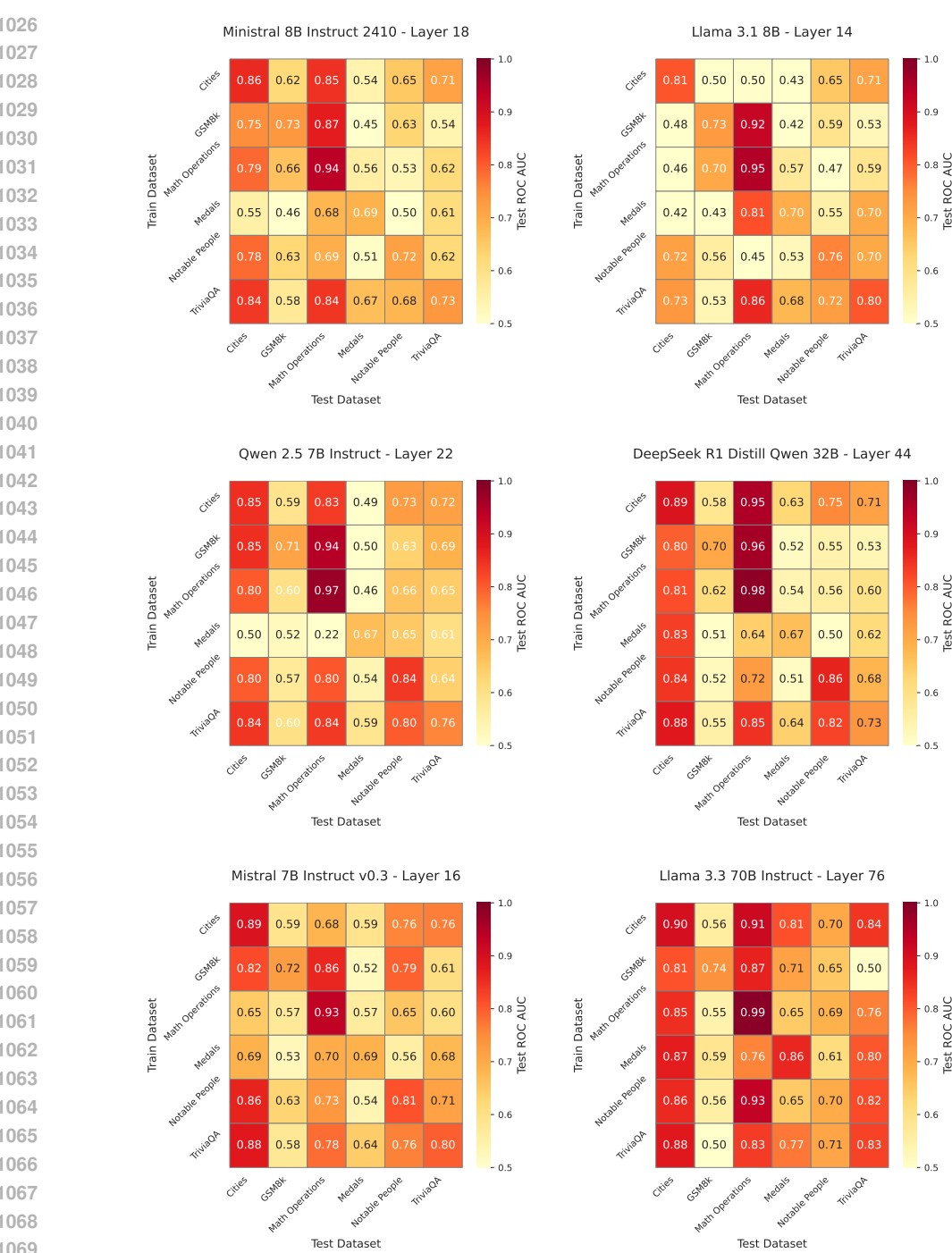

Figure 6: AUROC scores on each dataset for the direction learned on each dataset individually for all models. Average AUROC over 5 folds is reported (Section 4.3).

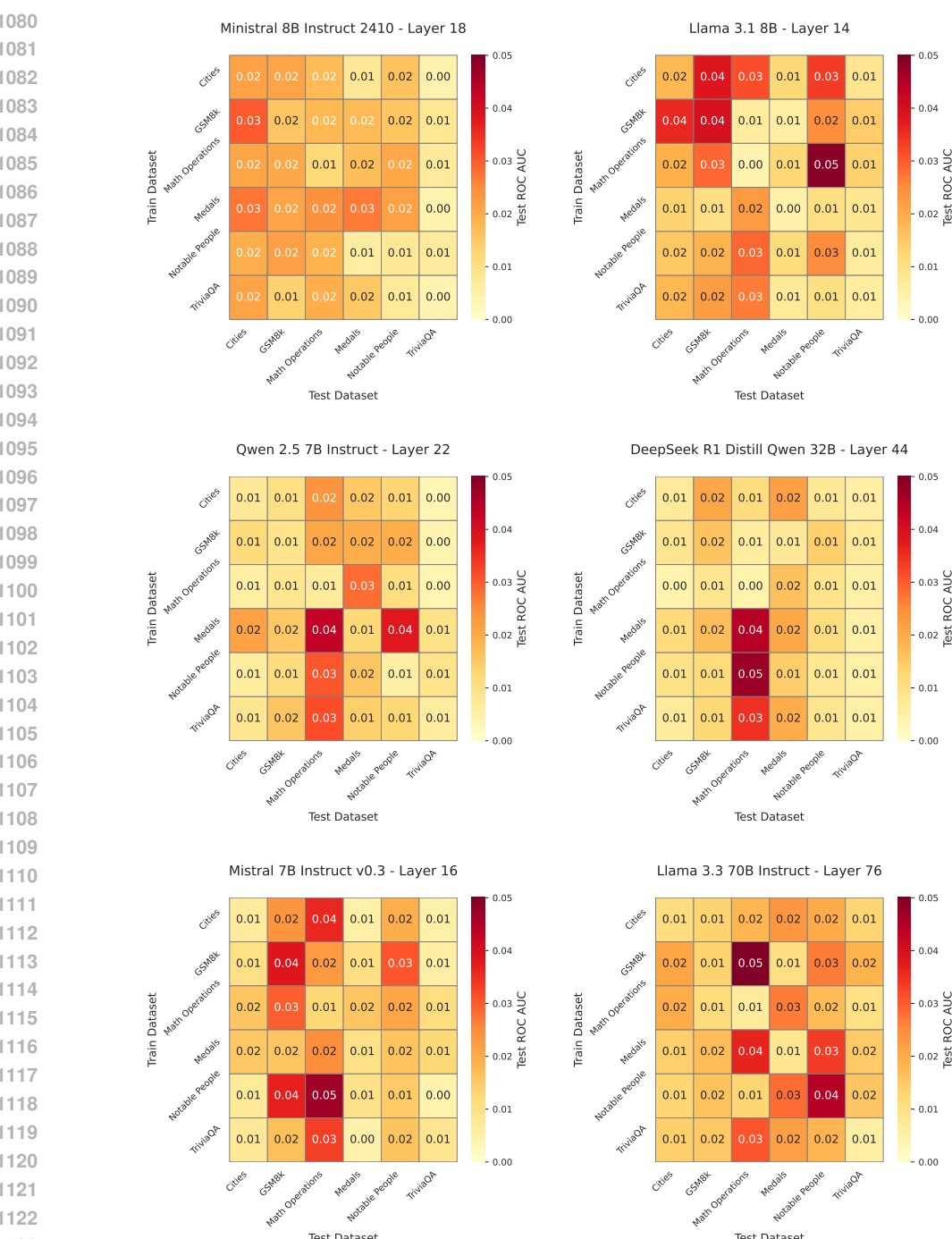

Figure 7: Standard deviations of the values in Figure 6, computed over 5 folds.

## B.6 COSINE SIMILARITIES

Figure 8 reports cosine similarities between the directions learned on the different datasets, for all models. Notice the cosine similarity ranges from -1 to +1, with 0 indicating orthogonality.

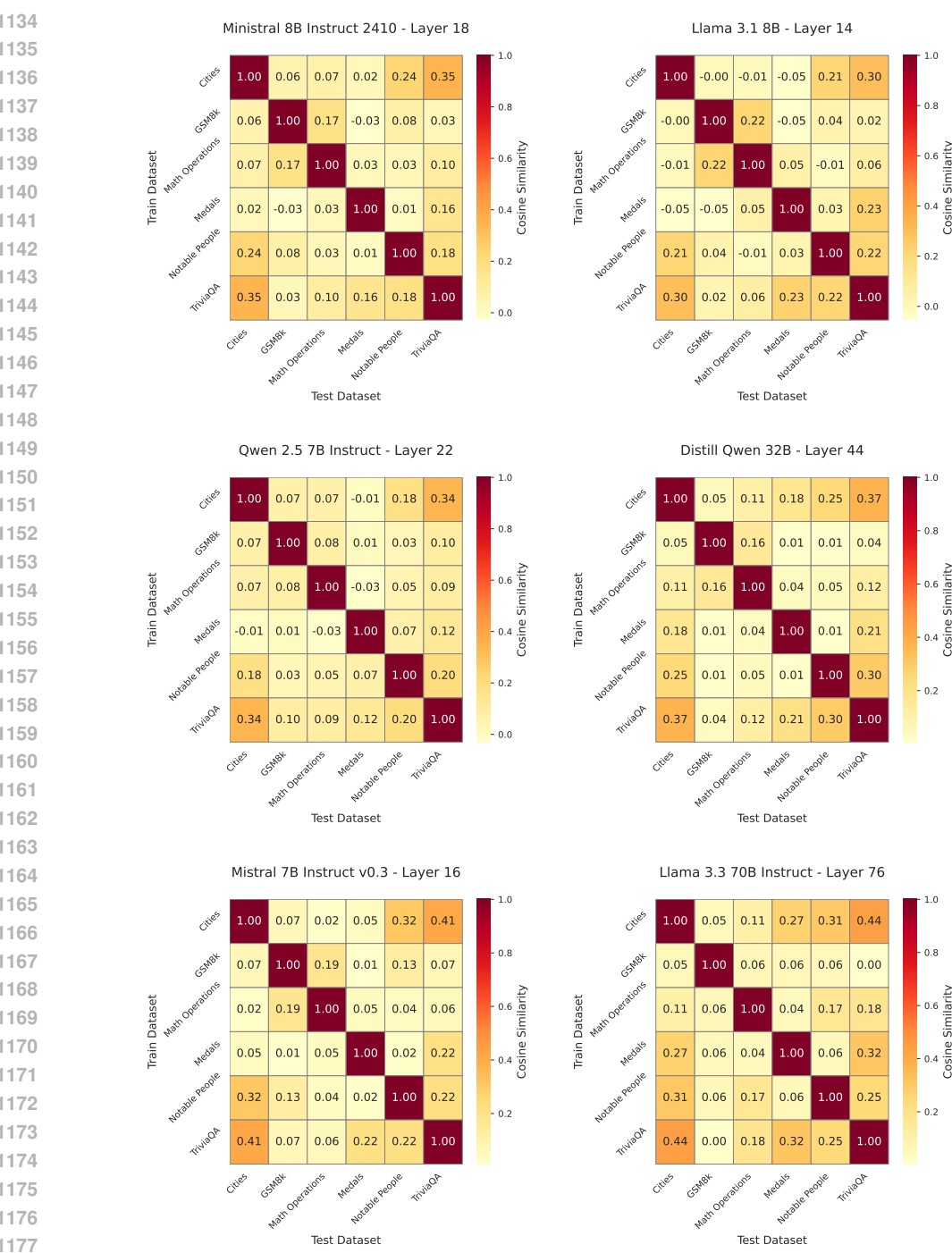

Figure 8: Cosine similarities for directions trained with different datasets. Following the same method as in Section 4.3, we average the directions over 5 folds and provide cosine similarities for these averages.

## B.7 CORRECTNESS DIRECTION PERFORMANCE ACROSS LAYERS

Figure 9 shows the in-distribution performance of the direction trained on each dataset over the layers of each considered model, complementing Figure 2.

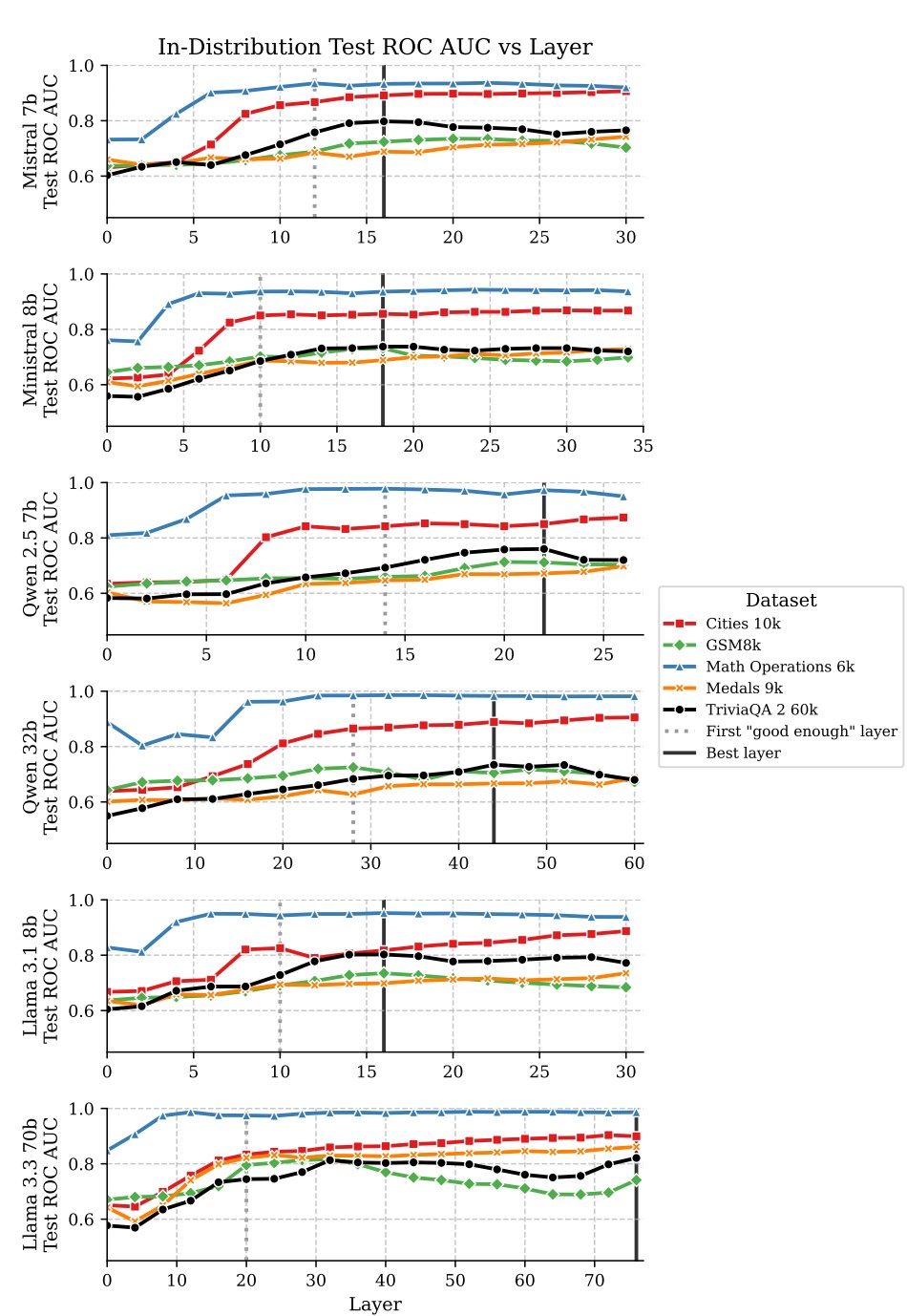

Figure 9: AUROC for each dataset, model and layer for the direction approach explained in Sec 3.2. The direction is trained and tested on the same dataset (using cross-validation). We collect activations every 2 layers for small (<10B parameters) models and every 4 layers for large (>10B parameters) models. The best layer is chosen as in Section 4.2, and the first "good enough" layer is the first layer that achieves 90% of the accuracy of the best layer in TriviaQA.

B.8 PROMPTS

Below we report the prompts used for our evaluations.

Prompt for Cities dataset:

I am going to ask you a question about cities. End your sentence with {eos_token}.Here are some examples of questions that might help you:
—
Question: In which country is the city of Barcelona located?
Answer: Spain{eos_token}
—
Question: In which country is the city of London located?
Answer: United Kingdom{eos_token}
—
Question: In which country is the city of Beijing located?
Answer: China{eos_token}
—
Question:{question}
Answer:

Prompt for Notable People dataset:

I am going to ask you what year a person was born. End your sentence with {eos_token}.Here are some examples of questions that might help you:
—
Question: What year was Barack Obama (politician from US) born?
Answer: 1961{eos_token}
—
Question: What year was Vladimir Putin (politician from Russia) born?
Answer: 1952{eos_token}
—
Question: What year was Xi Jinping (politician from China) born?
Answer: 1953{eos_token}
—
Question:{question}
Answer:

Prompt for Medals dataset:

I am going to ask you a question about the Olympics. End your sentence with {eos_token}.Here are some examples of questions that might help you:
—
Question: Which country won gold in Gymnastics Women's Team All-Around in the 1928 Summer Olympics?
Answer: Netherlands{eos_token}
—
Question: Which country won gold in Hockey Women's Hockey in the 2004 Summer Olympics?
Answer: Germany{eos_token}
—
Question: Which country won gold in Fencing Men's Sabre, Individual in the 1964 Summer Olympics?
Answer: Hungary{eos_token}
—
Question:{question}
Answer:

Prompt for TriviaQA dataset:

I am going to ask you a question. Answer concisely. End your sentence with `{eos_token}`.Here are some examples of questions that might help you:
—
Question: In which month are St David's Day and St Patrick's Day celebrated in the UK?
Answer: March`{eos_token}`
—
Question: What is the common English name of Mozart's Serenade for Strings in d major?
Answer: A little night music`{eos_token}`
—
Question: In which US State do teams play baseball in the Cactus League?
Answer: Arizona`{eos_token}`
—
Question:`{question}`
Answer:

Prompt for Math Operations dataset:

I am going to ask you questions about maths. Answer with an integer value, without decimal places. End your sentence with `{eos_token}`.Here are some examples of questions that might help you:
—
Question: What is 604 minus 866?
Answer: -262`{eos_token}`
—
Question: What is 927 plus 855?
Answer: 1782`{eos_token}`
—
Question: What is 531 times 955?
Answer: 507105`{eos_token}`
—
Question:`{question}`
Answer:

Prompt for GSM8K dataset:

I am going to ask you a question that requires your answer in a boxed integer. End your sentence with `{eos_token}`.Here are some examples of questions that might help you:
—
Question: Weng earns \$12 an hour for babysitting. Yesterday, she just did 50 minutes of babysitting. How much did she earn?
Answer: $\boxed{10}$`{eos_token}`
—
Question: Julie is reading a 120-page book. Yesterday, she was able to read 12 pages and today, she read twice as many pages as yesterday. If she wants to read half of the remaining pages tomorrow, how many pages should she read?
Answer: $\boxed{42}$`{eos_token}`
—
Question: Mark has a garden with flowers. He planted plants of three different colors in it. Ten of them are yellow, and there are 80% more of those in purple. There are only 25% as many green flowers as there are yellow and purple flowers. How many flowers does Mark have in his garden?
Answer: $\boxed{35}$`{eos_token}`
—
Question:`{question}`
Answer:

Prompt for the verbalized confidence experiment:

I am going to ask you about your confidence to answer a question. The confidence indicates how likely you think your answer will be true. Please respond with only a percentage and end with {eos_token}, so your answer should be following the format

Answer: (percentage)%{eos_token}

How confident are you that you can answer correctly '{question}'? Answer:

## C   THE USE OF LARGE LANGUAGE MODELS IN THIS RESEARCH PAPER

Besides being the subject of the investigation, the authors acknowledge having used Large Language Models in polishing the writing of some sections and for finding related works to be mentioned.

