# OpenReview forum: "No Answer Needed: Predicting LLM Answer Accuracy from Question-Only Linear Probes"
_ICLR.cc/2026/Conference — Submitted to ICLR 2026_

### Official Review · Reviewer_vgab · 2025-10-27

**Soundness:** 3
**Presentation:** 3
**Contribution:** 2
**Rating:** 4
**Confidence:** 4

**Summary:**

This work proposes a simple method which identifies a ``correctness'' vector as the difference between the average representations of incorrect and correct answers, projecting new problems' representations onto the computed vector to extract the LLMs learned evaluation of its own answer's correctness on the problem.

**Strengths:**

1. The paper includes sufficient detail to understand the complete experimental and methodological design
2. The text is in general clearly written.

**Weaknesses:**

1. Experimental setting: datasets used are mostly synthetic and private. Weak justification was provided for the exclusion of the typically included datasets.
2. Lack of clarity in the description of experiments and results. Figures 3 and 4, in particular, are lacking. Figure 3 is difficult to interpret without an idea of how the baselines perform. Figure 4 is difficult to understand given the hazy decision rule. Table 3 also has some strange values which go unaddressed. Table 3 would benefit greatly from a baseline LLM performance on the immediate task (or a numerically equivalent methodological baseline which naively predicts ``Correct'' at all times).
3. Lack of explicit motivation or downstream design: the text does not go far to motivate the work. In the same vein, the experimental design is such that the method's downstream utility is hard to gauge: a decision rule, or a process for finding a decision rule, is not described. Instead, results are aggregated for all possible thresholding rules. Correctness prediction is a 2nd-degree task, in that it is a prediction on a prediction. However, its utility in improving its 1st-degree task (Question answering) is not discussed.

**Questions:**

1. Figure 4 demonstrates the distributions of your three classes on their correctness scores. We see that, particularly in the 7B case, "Wrong" is entirely overlapped by the other two classes. This means that, given this observation, for any correctness score, one of the other classes will have a higher probability. This means that the optimal maximum likelihood estimation-based prediction would never predict "wrong". In practice, I understand that you are in fact picking a naive score threshold (I suppose two thresholds since you've defined a third ``IDK'' class), and are measuring AUROC over all thresholds. This makes Figure 4 seem strange, however, since score overlap across classes should not be possible with a thresholding decision. Can you comment on the class-distribution interpretation of Figure 4, and clarify how the data in this Figure was compiled?
2. Some of the datasets you use have very high direct task accuracy on modern LLMs. GSM8K, for example, is typically considered too easy a dataset at this point. I would also expect Cities' accuracy to be extremely high on the immediate task, assuming you only used major cities to construct questions. For these datasets, always predicting ``True'' would appear as having very high accuracy. In fact, such a prediction scheme would allow us to always match downstream performance in the correctness classification task. How do your own reported statistics compare with the direct tasks'? For GSM8K, the only public dataset you use, it seems likely to be much weaker. In your main table, you should indicate vanilla-prompted direct-task performance (or the naive classifier performance we get by always predicting ``True'').
3. In truth, semantically, all single-concept free-form generation problems are effectively multiple choice. This is because in practice, when temperature-sampled (i.e. self-consistency), the LLM will only generate a handful of potential answers. Most of the time, with the exception of very hard datasets, one of the LLMs potential answers will be the correct one. This commonly observed behavior makes your dataset-selection justification hard to buy. In addition, even if there were strong random-selection bias, such a bias would be present in each method's observed performance and so the comparison would remain fair. In fact, this would probably be a good test for training robustness. If 5% of the collected ``True'' datapoints were lucky guesses, would this really impact training substantially? Besides, you can always take public datasets and convert them from multiple-choice to free-form generation by simply not providing the multiple choices in your prompt. Can you either provide experimental results on a few new datasets or provide stronger justification of your experimental design?
4. You present in Figure 3 your method's performance when trained on datasets other than TriviaQA. How does this compare to the assessor?
5. You have a massive volume of training data, which is all high-dimensional. Why not use a standard MLP classifier as an assessor instead of logistic regression? This seems the perfect setting for a softmaxxed MLP.
6. While the work is not expressly motivated in the text, motivation for such a correctness prediction seems clear: we can estimate answer-verification a priori in order to develop schemes to improve post-training performance. While I understand that this verification scheme is not in scope, it would be trivial to test your method's downstream utility on a binary task such as True-or-False questions, where if the classification is categorized by your method as incorrect then the true answer must be the single alternative class. Effectively, you could procedurally generate the ``True'' class for each question and test whether this answer is correct, making your method immediately applicable to the task. I'd like to see this on something like FOLIO, which is True/Unknown/False (you can use your "IDK" for maybe, or just cut out all the "Unknown"-labelled problems from the dataset).
7. In Table 3, you often report sub-random performances (less than 0.5). What is happening in the models to produce such class inversions?
8. I don't understand why you don't commit to a single decision boundary. In training, the identification of an optimal boundary should be simple to find. In truth, this method is unusable without committing to a boundary, and your experimental results are all weakened and made less interpretable by requiring an AUROC measurement.

---

> ### Author Response · Authors · 2025-11-21
> **Rebuttal 1/5**
>
> We thank the reviewer for their attention to detail. Below we respond to each *comment/weakness*, and hope that you can reconsider your score in light of our discussion.
>
> *Experimental setting: datasets used are mostly synthetic and private. Weak justification was provided for the exclusion of the typically included datasets.*
>
> All datasets we use are either public benchmarks (TriviaQA, GSM8K) or are generated from public, CC-licensed tables (Cities, Notable People, Medals, Math Operations; Table 1 and Sec. 4.1.1). We would like to clarify that none of them are private, and they were chosen them because they (i) allow open-ended answers with fully automatic grading, (ii) give mid-range accuracy for our models (Table 5), so both correct and incorrect classes are reasonably well populated, and (iii) attempt to reduce exact benchmark memorisation by using our own templates over public sources.
>
> *Lack of clarity in the description of experiments and results. Figures 3 and 4, in particular, are lacking. Figure 3 is difficult to interpret without an idea of how the baselines perform. Figure 4 is difficult to understand given the hazy decision rule. Table 3 also has some strange values which go unaddressed. Table 3 would benefit greatly from a baseline LLM performance on the immediate task (or a numerically equivalent methodological baseline which naively predicts ``Correct'' at all times).*
>
> We warmly thank the reviewer for the suggestions and agree that some figures can be made clearer. To clarify: Figure 3 reports only the performance of our correctness direction, and the corresponding assessor baselines (trained on TriviaQA) are already in Table 3, with extended results in Appendix B.3. We will make the Figure 3 caption more self-contained and explicitly cross-reference these tables. We also plan to add an assessor heatmap (assessors trained per dataset, mirroring Fig. 3) in the appendix, which will take some time. Figure 4 is purely descriptive: it shows the histogram of scalar scores (projections on the correctness direction) for three types of answers (“right”, “wrong”, “IDK”), not the outputs of a three-class classifier. Overlap between these histograms is therefore expected and compatible with a single-threshold decision rule. We will clarify this in the caption, emphasising that the point of the plot is to show that “IDK” answers concentrate at the lowest scores, supporting the “confidence” interpretation. For Table 3, we will (i) add a short explanation about entries that are close to random performance (around 0.5), (ii) bring up direct task accuracies from Table 5 in the main text near Table 3, and (iii) include, in the appendix, a simple “always correct” baseline and a fixed-threshold version of our method for ease of comparison.
>
> *Lack of explicit motivation or downstream design: the text does not go far to motivate the work. In the same vein, the experimental design is such that the method's downstream utility is hard to gauge: a decision rule, or a process for finding a decision rule, is not described. Instead, results are aggregated for all possible thresholding rules. Correctness prediction is a 2nd-degree task, in that it is a prediction on a prediction. However, its utility in improving its 1st-degree task (Question answering) is not discussed.*
>
> We thank this reviewer's comment on the framing and theory of change of our paper. In its current form, our primary goal is scientific: we want to test whether a linearly accessible, pre-generation correctness signal exists in LLM internals, how strong it is, and how it generalises across models and datasets. This is somewhat different from proposing a complete deployment pipeline, as we expect any practical system to combine multiple signals (e.g., internal probes, sampling-based methods, verbalised confidence) rather than rely on a single probe. Rather, we see it more as sketching out the contours, limitations, and mechanisms for a possible step to include in such a pipeline, which may have utility in improving the 1st-degree task.
>
> AUROC is standard in classicfication and probing work and directly answers our core question about separability without committing to any particular application-specific operating point. Finding a decision rule is a simple matter of choosing the threshold that maximises accuracy on a dataset. We will state this more explicitly in the introduction and conclusion, and we have also included accuracy results for a fixed decision rule in Table 7.
>
> [Cont. below]

---

> ### Author Response · Authors · 2025-11-21
> **Rebuttal 2/5**
>
> *Figure 4 demonstrates the distributions of your three classes on their correctness scores. We see that, particularly in the 7B case, "Wrong" is entirely overlapped by the other two classes. This means that, given this observation, for any correctness score, one of the other classes will have a higher probability. This means that the optimal maximum likelihood estimation-based prediction would never predict "wrong". In practice, I understand that you are in fact picking a naive score threshold (I suppose two thresholds since you've defined a third ``IDK'' class), and are measuring AUROC over all thresholds. This makes Figure 4 seem strange, however, since score overlap across classes should not be possible with a thresholding decision. Can you comment on the class-distribution interpretation of Figure 4, and clarify how the data in this Figure was compiled?*
>
> We apologise for the confusion. Figure 4 does not depict the output of a three-class classifier. For each model and dataset, we (i) train a binary direction on “correct” vs “not correct”, treating “I don’t know” as incorrect, and then (ii) plot the scalar scores for those examples, grouped by their actual answer types (“right”, “wrong”, “IDK”). The overlap across the three histograms simply reflects that the binary score is not perfectly discriminative, which is consistent with the non-perfect AUROCs we report. The figure is intended to show that “IDK” answers cluster at the low-score tail, which supports the view of this direction as a confidence axis, not to define a 3-way decision rule. We will clarify this in the caption and main text.
>
> *Some of the datasets you use have very high direct task accuracy on modern LLMs. [...]*
>
> For the open models we use, accuracies are far from saturated. As shown in Table 5, GSM8K accuracy ranges from 10.7% to 44.3%, and Cities from 45.8% to 80.3%, depending on the model, so there is substantial room for both correct and incorrect examples in all datasets. We will move key numbers from Table 5 into the main text near Table 3 to make this clearer. A naive “always correct” classifier has AUROC = 0.5 by definition, and our correctness directions exceed this on all factual datasets (TriviaQA, Notable People, Cities, Math Operations, Medals) across all models (Table 3).
>
> In the appendix, we provide a new Table 7 (in App. B.3), which reports the accuracy of a thresholded classifier based on the correctness direction, both with a TriviaQA-trained threshold and with dataset-specific thresholds. This makes the comparison more concrete: on TriviaQA, for example, our classifier attains accuracies around 0.67–0.75, while an “always correct” baseline would match the underlying QA accuracy but still have AUROC 0.5. We therefore see AUROC as the right primary metric for our scientific question but that reasonable decision rules are available when one wants to trade off errors in practice!

---

> > ### Author Response · Authors · 2025-11-21
> > **Rebuttal 3/5**
> >
> > Table 7. Accuracy of a classifier based on the correctness direction at the optimal layer. For each test dataset, we display results for the direction trained on TriviaQA and for the direction trained on the dataset itself. The threshold for the classifier is the optimal one for the training data.
> > | Model            | TriviaQA (TQA train) | TriviaQA (self train) | N. people (TQA train) | N. people (self train) | Cities (TQA train) | Cities (self train) | Math ops. (TQA train) | Math ops. (self train) | Medals (TQA train) | Medals (self train) | GSM8K (TQA train) | GSM8K (self train) |
> > |------------------|----------------------|------------------------|------------------------|-------------------------|--------------------|----------------------|------------------------|-------------------------|--------------------|----------------------|--------------------|----------------------|
> > | Llama 3.1 8B     | 0.728                | 0.728                  | 0.522                  | 0.688                   | 0.551              | 0.714                | 0.755                  | 0.874                   | 0.612              | 0.682                | 0.500              | 0.670                |
> > | Llama 3.3 70B    | 0.750                | 0.750                  | 0.621                  | 0.651                   | 0.797              | 0.808                | 0.520                  | 0.943                   | 0.553              | 0.782                | 0.500              | 0.671                |
> > | Qwen 2.5 7B      | 0.688                | 0.688                  | 0.639                  | 0.764                   | 0.515              | 0.762                | 0.792                  | 0.938                   | 0.512              | 0.640                | 0.501              | 0.654                |
> > | Deepseek R1 32B  | 0.671                | 0.671                  | 0.588                  | 0.793                   | 0.612              | 0.802                | 0.848                  | 0.961                   | 0.519              | 0.636                | 0.500              | 0.660                |
> > | Mistral 7B       | 0.721                | 0.721                  | 0.523                  | 0.724                   | 0.771              | 0.811                | 0.724                  | 0.858                   | 0.545              | 0.661                | 0.503              | 0.659                |
> > | Ministral 8B     | 0.670                | 0.670                  | 0.514                  | 0.647                   | 0.500              | 0.774                | 0.764                  | 0.830                   | 0.532              | 0.656                | 0.502              | 0.671                |
> >
> > *In truth, semantically, all single-concept free-form generation problems are effectively multiple choice. This is because in practice, when temperature-sampled (i.e. self-consistency), the LLM will only generate a handful of potential answers. Most of the time, with the exception of very hard datasets, one of the LLMs potential answers will be the correct one. This commonly observed behavior makes your dataset-selection justification hard to buy. In addition, even if there were strong random-selection bias, such a bias would be present in each method's observed performance and so the comparison would remain fair. In fact, this would probably be a good test for training robustness. If 5% of the collected ``True'' datapoints were lucky guesses, would this really impact training substantially? Besides, you can always take public datasets and convert them from multiple-choice to free-form generation by simply not providing the multiple choices in your prompt. Can you either provide experimental results on a few new datasets or provide stronger justification of your experimental design?*
> >
> > We agree that, at inference time, an LLM’s support over answers is finite and often concentrated on a handful of plausible candidates. However, this is conceptually distinct from the evaluation format being multiple choice. Multiple-choice evaluation induces substantial chance correctness (e.g., 25% with four options), which directly biases the centroids used to define the direction. This is precisely why we avoid MCQ formats in Sec. 4.1.1 and use open-ended questions with bounded but broad answer sets (e.g., years, countries) and exact, automatic grading. Our custom datasets are built from public tables with our own templating, which both reduces the impact of random guessing and mitigates benchmark-specific memorisation. We will state this rationale more clearly and note that converting existing MCQ benchmarks into free-form format, as the reviewer suggests, is a natural direction for future work.

---

> > > ### Author Response · Authors · 2025-11-21
> > > **Rebuttal 4/5**
> > >
> > > *You present in Figure 3 your method's performance when trained on datasets other than TriviaQA. How does this compare to the assessor?	In the main text, Table 3 and Appendix B.3 compare the direction trained on TriviaQA to assessors also trained on TriviaQA, showing that assessors tend to perform slightly better in-distribution but generalise more poorly out of distribution.*
> > >
> > > Following the reviewer’s suggestion, we will add an appendix experiment where we also train assessors on each dataset individually (mirroring Fig. 3). This will take some time, as the it will require processing >200 GB of data again so we are queueing up the servers required.
> > >
> > > *You have a massive volume of training data, which is all high-dimensional. Why not use a standard MLP classifier as an assessor instead of logistic regression? This seems the perfect setting for a softmaxxed MLP.*
> > >
> > > Our main question is whether a linear internal signal is already sufficient to encode correctness/confidence in advance. For non-linear baselines we already include XGBoost assessors (Appendix B.3). These gain an extra 3–5 AUROC points in-distribution on TriviaQA but lose 4–7 points out-of-distribution, a clear sign of overfitting. We believe that an MLP assessor would further increase capacity but not significantly change the interpretability question, so we chose to keep the baseline set simple and focused. We hope this clears up the motivation, and frames our contribution within the hopes of many other mechanistic interpretability works which aim to approach this high-dimensional data as a space containing high-level concepts which can be represented linearly as directions (LRH).
> > >
> > > *While the work is not expressly motivated in the text, motivation for such a correctness prediction seems clear: we can estimate answer-verification a priori in order to develop schemes to improve post-training performance. While I understand that this verification scheme is not in scope, it would be trivial to test your method's downstream utility on a binary task such as True-or-False questions, where if the classification is categorized by your method as incorrect then the true answer must be the single alternative class. Effectively, you could procedurally generate the ``True'' class for each question and test whether this answer is correct, making your method immediately applicable to the task. I'd like to see this on something like FOLIO, which is True/Unknown/False (you can use your "IDK" for maybe, or just cut out all the "Unknown"-labelled problems from the dataset).*
> > >
> > > We agree that applying the probe in a true/false setting such as FOLIO is an interesting direction. Doing so, however, requires extra design choices (how to construct the alternative class, how to integrate with the dataset, how to choose thresholds) and additional compute. In this paper we focus on characterising the internal correctness signal itself (where it appears in the network, how strong it is, and how it generalises, and so on) rather than optimising a full verification pipeline. We will expand the discussion section to emphasise that we expect practical systems to combine our probe with other signals rather than rely on it alone!
> > >
> > > *In Table 3, you often report sub-random performances (less than 0.5). What is happening in the models to produce such class inversions?*
> > >
> > > In Table 3, the few AUROC values below 0.5 are all close to random (typically 0.45–0.50) and arise in settings where the signal is weak (notably some assessor scores on Math Operations and GSM8K, and one GSM8K value for our direction on Llama 3.3 70B). For our direction, AUROC < 0.5 simply means that, for that model–dataset pair, the learned scalar score is mildly anti-correlated with correctness. We suspect this can happen because (i) the underlying signal is very weak (so the noise from finite samples), and/or (ii) the direction, which is only defined up to sign, ends up slightly “flipped” relative to how we would orient it if we optimised sign per dataset. We very carefully do not post-hoc flip the sign per dataset, to avoid any cherry-picking and to keep the analysis comparable across settings. We will add a short explanation of this behaviour to the text.

---

> ### Author Response · Authors · 2025-11-21
> **Rebuttal 5/5**
>
> *I don't understand why you don't commit to a single decision boundary. In training, the identification of an optimal boundary should be simple to find. In truth, this method is unusable without committing to a boundary, and your experimental results are all weakened and made less interpretable by requiring an AUROC measurement.*
>
> We use AUROC because it is threshold-free, widely used in probing work, and directly measures how linearly separable the two classes are, which is our main scientific question. In any deployment, a practitioner would choose a decision boundary using a small validation set, tuned to some preferred trade-off between false positives and false negatives. To address the reviewer’s concern, we have added an appendix table (new Table 7, in Appendix B.3) that commits to concrete thresholds, where for each model and training dataset, we (i) choose a single scalar threshold that maximises accuracy on that training dataset, and (ii) report the resulting classifier accuracy across all test datasets. The “TriviaQA” columns show the case where the threshold and direction are both trained on TriviaQA and then transferred to other datasets, while the “Itself” columns show the case where both the direction and the threshold are trained on the target dataset.
>
> What happens when you do this? In-distribution, a simple threshold on the correctness direction yields pretty sensible accuracies (e.g., around 0.67–0.75 on TriviaQA, and up to around 0.83–0.96 on the synthetic factual datasets), which is around the AUROCs you see in Table 3. Then OOD, when we transfer the TriviaQA-trained threshold to other datasets, accuracies fall in line with the cross-dataset AUROCs: they remain above random for other factual datasets, but stay around chance on GSM8K (~0.50), which reinforces our conclusion that the question-only correctness signal does not transfer well to multi-step reasoning. We hope this gives confidence that (i) it is straightforward to instantiate our probe as a thresholded classifier with a single decision boundary, and (ii) AUROC is still the more informative metric for our purposes, because it describes the underlying correctness signal independent of class imbalance and threshold calibration.

---

> ### Comment · Reviewer_vgab · 2025-11-21
> **Response to Author Rebuttal (part 1)**
>
> *All datasets we use are either public benchmarks (TriviaQA, GSM8K) or are generated from public, CC-licensed tables (Cities, Notable People, Medals, Math Operations; Table 1 and Sec. 4.1.1). We would like to clarify that none of them are private, and they were chosen them because they (i) allow open-ended answers with fully automatic grading, (ii) give mid-range accuracy for our models (Table 5), so both correct and incorrect classes are reasonably well populated, and (iii) attempt to reduce exact benchmark memorisation by using our own templates over public sources.*
>
> While the source data might itself be public, I have no way of knowing for sure exactly which questions make up your synthetic datasets. If I were to to attempt to reproduce your methods, and found that I was getting different empirical results on experiments I attempt to reproduce from your work, I would not know if it is due to an error in the reproduction of the method or simply the fact that the data is different. While true that there is a risk of memorization when using standardized datasets, it seems that the literature by-and-large has not managed to identify cases of serious empirical inconsistencies stemming from such a phenomenon. It seems that typically, methods’ reported results on open-source and potentially leaked data are, rank-wise, similar to the reporting on synthetic or closed-source data. So, I’m not sure how valid this justification is for diverging from the literature in terms of benchmark settings.
>
> *We warmly thank the reviewer for the suggestions and agree that some figures can be made clearer. To clarify: Figure 3 reports only the performance of our correctness direction, and the corresponding assessor baselines (trained on TriviaQA) are already in Table 3, with extended results in Appendix B.3. We will make the Figure 3 caption more self-contained and explicitly cross-reference these tables. We also plan to add an assessor heatmap (assessors trained per dataset, mirroring Fig. 3) in the appendix, which will take some time.*
>
> The information conveyed in Figure 3 is clear. What is unclear is the message we are meant to draw from it. The in-text discussion you provide describes the interpretation of the figure as (1) showing that generally training one one dataset does not generalize well to other datasets, and (2) that all datasets lead to “decent” in-distribution performance. The second claim you make based on Figure 3 requires comparisons to baselines to confirm, since “decent” is a subjective and relative qualification. Considering that there are 36 boxes per heatmap, the reader can not be expected to scroll back and forth between the figure and the appendix 36 times to compare each box.  Why can’t you include two numbers in each box, one for the Assessor baseline and one for your method?
>
> *For Table 3, we will (i) add a short explanation about entries that are close to random performance (around 0.5), (ii) bring up direct task accuracies from Table 5 in the main text near Table 3, and (iii) include, in the appendix, a simple “always correct” baseline and a fixed-threshold version of our method for ease of comparison.*
> While this does not explicitly address directly my concern with Figure 3, these changes seem beneficial to the paper. Since in ICLR authors can upload amended versions of their papers for reviewers, there is no need to promise future changes to the paper, unless they are in the form of experiments which take more than three weeks to run.
>
> *Figure 4 is purely descriptive: it shows the histogram of scalar scores (projections on the correctness direction) for three types of answers (“right”, “wrong”, “IDK”), not the outputs of a three-class classifier. Overlap between these histograms is therefore expected and compatible with a single-threshold decision rule. We will clarify this in the caption, emphasising that the point of the plot is to show that “IDK” answers concentrate at the lowest scores, supporting the “confidence” interpretation.*
>
> This clarifies things. In this case, should the score distributions not be identical (or close to identical), since the correctness of “right” if it is indeed the correct answer is no different than the correctness of “wrong” if it is indeed the wrong answer? Also, you argue that “I don’t know” is indicative of the model internally knowing with confidence that it is wrong. However, is “I don’t know” not meant to be indicative of simply a low confidence prediction? Is being very small on the correctness direction not indicative of incorrectness? If I choose “right” and then feel confident that I’ve chosen incorrectly, why would I not just choose “wrong” then?

---

> ### Comment · Reviewer_vgab · 2025-11-21
> **Response to Author Rebuttal (part 2)**
>
> *In its current form, our primary goal is scientific: we want to test whether a linearly accessible, pre-generation correctness signal exists in LLM internals, how strong it is, and how it generalizes across models and datasets.*
> In your work, you state that your goals are: “In this work, we investigate the complementary scientific question of whether an LLM’s residual stream activations—captured immediately after it processes a query—contain a latent signal that predicts if its eventual output will be correct”, and in particular, we are interested in studying if the signal we identify indicates future performance without relying on dataset-specific surious features, rather than optimising predictive  power for specific deployment scenarios.” However, previous work has already found that LLM’s hidden states/residual streams contain latent confidence estimates. You cite some of these works in your paper. The research question of investigating whether there is out-of-distribution generalization (“without relying on dataset-specific spurious features”) is indeed a research question that you answer. However, neither of these questions require the design of a novel methodology. Your results on only the assessor indicate that some generalization occurs from TriviaQA to your other datasets. To this end, why do you emphasize that your novel method is more accurate than the alternative approaches you identify? Since you make no explicit hypothesis or research question connected to the specific design of your method, the implicit hypothesis which we see in most work is that your hypothesis is that your novel method will perform better than existing work. Again, without a specifically stated purpose for the testing of this, the implicit motivation for such an investigation is that it is useful to have performant methods for this problem setting. So, indicating usefulness goes a long way to motivate the work.
>
> *For the open models we use, accuracies are far from saturated. As shown in Table 5, GSM8K accuracy ranges from 10.7% to 44.3%, and Cities from 45.8% to 80.3%, depending on the model, so there is substantial room for both correct and incorrect examples in all datasets. We will move key numbers from Table 5 into the main text near Table 3 to make this clearer. A naive “always correct” classifier has AUROC = 0.5 by definition, and our correctness directions exceed this on all factual datasets (TriviaQA, Notable People, Cities, Math Operations, Medals) across all models (Table 3).*
> OK. Llama3.1 8B is reported as having ~80\% accuracy on GSM8K with few-shot examples and COT. I suppose you are doing neither COT nor few-shot prompting, which is fine since the goal is to build a dataset of answers to classify. With that said, it’s conceivable that COT and few-shot simply serve to ensure that the autoregressive generation is aligned with what we expect to see, in which case even when not using COT or few-shot, the model’s internal understanding of the problem is rich despite its low accuracy on the task. Ideally, datasets on which the LLM struggles even when the standard prompt augmentations are used, would be more indicative of whether the internal representation encodes correctness even when it does not encode the correct answer.
>
> *In the appendix, we provide a new Table 7 (in App. B.3), which reports the accuracy of a thresholded classifier based on the correctness direction, both with a TriviaQA-trained threshold and with dataset-specific thresholds. This makes the comparison more concrete: on TriviaQA, for example, our classifier attains accuracies around 0.67–0.75, while an “always correct” baseline would match the underlying QA accuracy but still have AUROC 0.5. We therefore see AUROC as the right primary metric for our scientific question but that reasonable decision rules are available when one wants to trade off errors in practice!*
> OK. Changes made to the paper should be indicated in the PDF in color.

---

> ### Comment · Reviewer_vgab · 2025-11-21
> **Response to Rebuttal (part 3)**
>
> *We agree that, at inference time, an LLM’s support over answers is finite and often concentrated on a handful of plausible candidates. However, this is conceptually distinct from the evaluation format being multiple choice. Multiple-choice evaluation induces substantial chance correctness (e.g., 25% with four options), which directly biases the centroids used to define the direction. This is precisely why we avoid MCQ formats in Sec. 4.1.1 and use open-ended questions with bounded but broad answer sets (e.g., years, countries) and exact, automatic grading. Our custom datasets are built from public tables with our own templating, which both reduces the impact of random guessing and mitigates benchmark-specific memorization. We will state this rationale more clearly and note that converting existing MCQ benchmarks into free-form format, as the reviewer suggests, is a natural direction for future work.*
> Your argument is only applicable in cases where the LLM’s free-form generation support would not include the correct choice (i.e. Pass@k is 0 for the given question). If the LLM’s generation support includes 4 options with uniform likelihood then the same random selection bias is introduced. I have not tested it, but I suspect that average Pass@k for the datasets you use would be very high, making your argument marginally applicable. Besides, you do not acknowledge the remainder of my comment, pointing out that the presence of such a bias in your training set would make for a strong assessment of your method’s training robustness, and that it remains a fair evaluation because the baseline methods which are trained suffer the same training biases.
>
> *Following the reviewer’s suggestion, we will add an appendix experiment where we also train assessors on each dataset individually (mirroring Fig. 3). This will take some time, as the it will require processing >200 GB of data again so we are queueing up the servers required.*
> OK.
>
> *Our main question is whether a linear internal signal is already sufficient to encode correctness/confidence in advance. For non-linear baselines we already include XGBoost assessors (Appendix B.3). These gain an extra 3–5 AUROC points in-distribution on TriviaQA but lose 4–7 points out-of-distribution, a clear sign of overfitting. We believe that an MLP assessor would further increase capacity but not significantly change the interpretability question, so we chose to keep the baseline set simple and focused. We hope this clears up the motivation, and frames our contribution within the hopes of many other mechanistic interpretability works which aim to approach this high-dimensional data as a space containing high-level concepts which can be represented linearly as directions (LRH).*
> I think that an MLP is simpler than XGBoost, so I don’t exactly buy that taking XGBoost over an MLP makes your baseline set simpler and more focused. This is a minor issue, however. More critically, you claim here, as above, that your main research question is whether the internal correctness signal is **linearly separable**. If this is indeed the main aim of your paper, it needs to be reflected in the introduction and the claims listed at the end of the introduction, and should probably be re-stated as a research question in your experimental setup. Also, I think it will affect the analysis you make of your experiments. For example, if the direction is linearly separable, how is it that regression algorithms or XGBoost are not able to learn the same predictions? What does it tell us about the nature of the internal signal that your method outperforms more expressive models? Again, I have a hard time accepting that your paper is purely scientific and exploratory, when the focus of your analysis and experimentation is simply to demonstrate that your method outperforms baselines.
>
> *We agree that applying the probe in a true/false setting such as FOLIO is an interesting direction. Doing so, however, requires extra design choices (how to construct the alternative class, how to integrate with the dataset, how to choose thresholds) and additional compute. In this paper we focus on characterising the internal correctness signal itself (where it appears in the network, how strong it is, and how it generalises, and so on) rather than optimising a full verification pipeline. We will expand the discussion section to emphasise that we expect practical systems to combine our probe with other signals rather than rely on it alone!*
> OK.

---

> ### Comment · Reviewer_vgab · 2025-11-21
> **Response to Rebuttal (part 4)**
>
> *n Table 3, the few AUROC values below 0.5 are all close to random (typically 0.45–0.50) and arise in settings where the signal is weak (notably some assessor scores on Math Operations and GSM8K, and one GSM8K value for our direction on Llama 3.3 70B). For our direction, AUROC < 0.5 simply means that, for that model–dataset pair, the learned scalar score is mildly anti-correlated with correctness. We suspect this can happen because (i) the underlying signal is very weak (so the noise from finite samples), and/or (ii) the direction, which is only defined up to sign, ends up slightly “flipped” relative to how we would orient it if we optimised sign per dataset. We very carefully do not post-hoc flip the sign per dataset, to avoid any cherry-picking and to keep the analysis comparable across settings. We will add a short explanation of this behaviour to the text.*
>
> Is (ii) not the result of (i)? It seems strange that random noise in the signal, regardless of its magnitude, would not average out given the size of your test sets. In Math Ops, where you have 6k samples, you report 0.4 AUROC on Qwen 2.5 7B. To me, such a result would indicate that the correctness direction for TriviaQA is somehow different from the correctness direction for Math. More abstractly, this would make sense if the underlying reasoning/verification systems were fundamentally different, which they are. Math and common-sense/knowledge reasoning are different both in terms of forward reasoning process and in terms of confidence evaluation / verification. This would, however, indicate that the signal is task-specific, and one could argue dataset-specific but that many datasets are overlapping in distribution.
>
> *We use AUROC because it is threshold-free, widely used in probing work, and directly measures how linearly separable the two classes are, which is our main scientific question. In any deployment, a practitioner would choose a decision boundary using a small validation set, tuned to some preferred trade-off between false positives and false negatives. To address the reviewer’s concern, we have added an appendix table (new Table 7, in Appendix B.3) that commits to concrete thresholds, where for each model and training dataset, we (i) choose a single scalar threshold that maximises accuracy on that training dataset, and (ii) report the resulting classifier accuracy across all test datasets. The “TriviaQA” columns show the case where the threshold and direction are both trained on TriviaQA and then transferred to other datasets, while the “Itself” columns show the case where both the direction and the threshold are trained on the target dataset.*
>
> The addition of this table is satisfactory. With that said, I will repeat myself here that if the aim of your work is specifically to evaluate the linear separability of correctness, this concept needs to be introduced throughout the text and much of your discussion may need changing. Currently, the specific goal of **linear separability** is not stated in the introduction or methodology, aside from describing the method itself as attempting to linearly separate out the signal. A preamble to your methodology explaining that you choose this methodology in order to check if it’s possible to find the signal as a linear separation would help. A claim in the bullet list of your introduction that you verify that the signal is linearly separable would help. A line in your introduction explaining that current methods for such correctness signals are perhaps more complex than necessary might be helpful. Re-phrasing your analysis to focus the linear separability and not the general superiority of the method on the task would help.

---

> ### Author Response · Authors · 2025-12-01
> **Rebuttal V2 (1/n)**
>
> > The addition of this table is satisfactory. With that said, I will repeat myself here that if the aim of your work is specifically to evaluate the linear separability of correctness, this concept needs to be introduced throughout the text and much of your discussion may need changing. [...]
>
> We thank the reviewer for the continued engagement and the highly constructive feedback regarding the framing of our work. We found your suggestion to reframe our paper to be very compelling, and have taken care to revise the paper accordingly. Below, we detail the specific changes and clarifications made in response to your latest comments.
>
> **Reframing the paper significantly.** We fully accept this critique. We have revised the manuscript to explicitly state that our primary research goal is testing the Linear Representation Hypothesis for correctness (verifying whether the signal is linearly separable in the residual stream) rather than solely proposing a new classification utility.
>
> Specific Revisions:
>
> Introduction: We explicitly state that we are testing the Linear Representation Hypothesis for self-correctness. We highlight that the existence of this linear separability is the primary finding. We have also rewritten the list of contributions to reflect this.
>
> Methodology: We added a short paragraph explaining that we chose a simple difference-of-means probe to test if the signal is accessible as a linear feature. If we used a complex non-linear probe, we could not make claims about linear separability.
>
> Conclusion Discussion: We rephrased the analysis to emphasize that the linear probe’s success confirms the existence of the signal, and have changed its tone accordingly.
>
> > I think that an MLP is simpler than XGBoost, so I don’t exactly buy that taking XGBoost over an MLP makes your baseline set simpler and more focused. This is a minor issue, however. More critically, you claim here, as above, that your main research question is whether the internal correctness signal is linearly separable. If this is indeed the main aim of your paper, it needs to be reflected in the introduction and the claims listed at the end of the introduction, and should probably be re-stated as a research question in your experimental setup. Also, I think it will affect the analysis you make of your experiments. For example, if the direction is linearly separable, how is it that regression algorithms or XGBoost are not able to learn the same predictions? What does it tell us about the nature of the internal signal that your method outperforms more expressive models? Again, I have a hard time accepting that your paper is purely scientific and exploratory, when the focus of your analysis and experimentation is simply to demonstrate that your method outperforms baselines.
>
> **Clarification on Baselines (Internal vs. External Inputs).** We realized there was a lack of clarity regarding the inputs used for the baselines versus our method, which we clarify now here:
>
> Our Method (Linear Probe): Operates on the LLM's internal residual stream activations.
>
> The Baselines (Assessors): Operate on OpenAI text embeddings of the question string (model-independent inputs).
>
> Therefore, the fact that our simple linear probe (on internals) outperforms the expressive XGBoost Assessor (on externals) proves that the internal activations contain specific ""self-knowledge"" or confidence information that is not present in the general semantic embedding of the question text, even when that text is analyzed by a (much more) powerful non-linear classifier. If XGBoost were trained on the internal activations, it would certainly learn the signal. Our point was more that the information gap between external semantics and internal states is not easily bridged by a significantly more expressive architecture. We think that a more expressive architecture applied to internal states would also learn the signal well, as shown by previous work.
>
> > To me, such a result [low/random AUC on Math] would indicate that the correctness direction for TriviaQA is somehow different from the correctness direction for Math.
>
> We agree with this interpretation. The signal is indeed task-specific in that regard: the "Factual Correctness" vector (which is shared across Trivia, Cities, People, and Medals) appears to be distinct or orthogonal to the "Arithmetic Correctness" vector.
>
> This reinforces the finding that the model's internal verification mechanisms for retrieval and reasoning are distinct. We have updated the discussion in Section 4.3.1 to reflect that this is likely a fundamental misalignment between the geometry of these different cognitive tasks, rather than only noise.

---

> > ### Author Response · Authors · 2025-12-01
> > **Rebuttal V2 (2/n)**
> >
> > > While the source data might itself be public, I have no way of knowing for sure exactly which questions make up your synthetic datasets. If I were to to attempt to reproduce your methods, and found that I was getting different empirical results on experiments I attempt to reproduce from your work, I would not know if it is due to an error in the reproduction of the method or simply the fact that the data is different.
> >
> > We thank the reviewer for this comment. We have now made the generated datasets available in the repo for the purposes of reproducing our work. We will also make stored activations available on academic torrents after deanonymization, which we hope provides confidence in our work being reproducible and contributing to the scientific understanding of LLMs.
> >
> > **On memorization.** The risk of memorisation is one of the motivations for having synthetic datasets, with the main motivation being the need to avoid multiple-choice tasks. Moreover, we are not establishing a new method to improve performance on existing benchmarks, in which case uniformity is essential; rather, we are studying a scientific question, so that we believe using a novel dataset is not detrimental. However, we concede the point on multiple choice generation below and acknowledge it as a limitation in our paper's methodology.
> >
> > > Your argument is only applicable in cases where the LLM’s free-form generation support would not include the correct choice (i.e. Pass@k is 0 for the given question). If the LLM’s generation support includes 4 options with uniform likelihood then the same random selection bias is introduced. I have not tested it, but I suspect that average Pass@k for the datasets you use would be very high, making your argument marginally applicable. Besides, you do not acknowledge the remainder of my comment, pointing out that the presence of such a bias in your training set would make for a strong assessment of your method’s training robustness, and that it remains a fair evaluation [...]
> >
> > You are right, and we accept this critique. Our hesitation to use multiple-choice datasets was mostly from a desire to minimize label noise for the centroid calculation, but we agree with your assessment: testing whether the probe can recover the signal despite the noise inherent in MCQ formats (or uniform-likelihood generation support) would indeed be a strong test of the method’s robustness. We concede that excluding these standard benchmarks weakened the evaluation of robustness, even if it cleaner for the initial signal discovery. We will acknowledge in the paper that the method’s applicability to MCQ/noisy-label settings is a limitation of the current study and a priority for future verification, rather than a fundamental incompatibility.
> >
> > > This clarifies things. In this case, should the score distributions not be identical (or close to identical), since the correctness of “right” if it is indeed the correct answer is no different than the correctness of “wrong” if it is indeed the wrong answer? Also, you argue that “I don’t know” is indicative of the model internally knowing with confidence that it is wrong. However, is “I don’t know” not meant to be indicative of simply a low confidence prediction? [...]
> >
> > "	Thanks for engaging deeply on this. We agree that our wording around “I don’t know” should emphasise low confidence rather than the model “knowing it is wrong with confidence,” in the self-awareness sense (like in recent literature). In the setup, the probe is trained only on a binary label indicating whether the eventual answer string exactly matches the ground truth (correct vs not), and learns a scalar score that separates these two sets. For any such score used with a single threshold, one should in general expect the distribution of scores on questions that end up correct to be shifted to the right of the distribution on questions that end up incorrect, the fact that both are graded against the same notion of correctness does not imply that these conditional score distributions must coincide. Figure 4 then further partitions the “incorrect” bucket post hoc according to the form of the generated string, using a pattern match to flag “I don’t know”, which are naturally treated as incorrect during training, and only separated in the plot to show that they concentrate at low scores. We will adjust the text to say that this is best interpreted as the correctness direction also tracking a low-confidence regime: being very negative on this direction is, by construction, indicative of the model predicting failure on that question, whether it goes on to output an IDK-like string or an incorrect guess. The human analogy (“if I choose ‘right’ and then feel confident I’ve chosen incorrectly, why not choose ‘wrong’?”) does not map directly to our setting, because the model never chooses between “right” and “wrong” labels, as those are assigned only by the grader, after the free-form answer has been produced.

---

> ### Author Response · Authors · 2025-12-01
> **Rebuttal V2 (3/3)**
>
> > OK. Llama3.1 8B is reported as having ~80% accuracy on GSM8K with few-shot examples and COT. I suppose you are doing neither COT nor few-shot prompting, which is fine since the goal is to build a dataset of answers to classify. With that said, it’s conceivable that COT and few-shot simply serve to ensure that the autoregressive generation is aligned with what we expect to see, in which case even when not using COT or few-shot, the model’s internal understanding of the problem is rich despite its low accuracy on the task. Ideally, datasets on which the LLM struggles even when the standard prompt augmentations are used, would be more indicative of whether the internal representation encodes correctness even when it does not encode the correct answer.
>
> We agree that CoT and few-shot can push GSM8K for our models much higher, and we use a simple, shared prompting scheme without CoT or few-shot across all datasets, so that question-only activations and probes are measured in a comparable regime. In that regime the open models we study are genuinely far from saturation (GSM8K 10–44% in Table 5), so the lack of transfer to GSM8K may be a property of this setting rather than a claim that the model never carries a richer internal solution. But this is important to us, as we are more interested in non-“elicitation-maxxed” regimes where the design space for getting the best out of a model is still open ended, since these are closer to many real deployments (and ones where piecing together if the model DOES know it can do well but doesn't is especially important), and for which we hope that scaling back the elicitation setup in GSM8K mimicks this reasonably well. Although, we agree that a natural extension is to repeat the analysis on tasks that remain hard even after strong prompt augmentation, to better separate “does not know” from “knows but is not yet elicited” and to be much closer to the frontier (one example that comes to mind is some type of "truesight" or author attribution task based on deep linguistic cues such as in Wen. et al 2025 and more broadly described by Gwern/Janus).
>
> Wen, J., Ankner, Z., Somani, A., Hase, P., Marks, S., Goldman-Wetzler, J., Petrini, L., Sleight, H., Burns, C., He, H. and Feng, S., 2025. Unsupervised Elicitation of Language Models. arXiv preprint arXiv:2506.10139.
>
> > The information conveyed in Figure 3 is clear. What is unclear is the message we are meant to draw from it. The in-text discussion you provide describes the interpretation of the figure as (1) showing that generally training one one dataset does not generalize well to other datasets, and (2) that all datasets lead to “decent” in-distribution performance. The second claim you make based on Figure 3 requires comparisons to baselines to confirm, since “decent” is a subjective and relative qualification. Considering that there are 36 boxes per heatmap, the reader can not be expected to scroll back and forth between the figure and the appendix 36 times to compare each box. Why can’t you include two numbers in each box, one for the Assessor baseline and one for your method?
>
> Thank you for this suggestion. We will comment again once this has been addressed (in the next few days), and included with the latest version of the paper.

---

### Official Review · Reviewer_xEzz · 2025-10-29

**Soundness:** 3
**Presentation:** 4
**Contribution:** 3
**Rating:** 6
**Confidence:** 3

**Summary:**

The paper tries to answer the question "Do LLMs know if they're gonna answer it right, before generating any response and after a question is read?" They extract residual stream activations after processing a question and train linear probes to predict answer correctness. They first identify the best layer (which is usually a middle layer) and then use train sets to identify the correctness vector in that layer. They show that training on TriviaQA generalizes to 4 other custom data. Their work mostly contributes to mechanistic interpretability by revealing that LLMs embed a latent correctness signal.

**Strengths:**

1. The empirical results on the evaluated benchmarks show that their method is substantially better than baselines and especially the confidence scores generated by the model itself.
2. They evaluate on a set of LLMs with varying sizes and architectures  (6 models from 3 families). The results are consistent across all models.
3. A qualitative analysis that provides more insights into error modes and why model's own confidence is not robust (e.g., A: 1961, Correct: 1962). The observation that responses containing "i don't know" are captured by their correctness direction further confirms the validity of the correctness direction.

**Weaknesses:**

Although I find the conclusions interesting and potentially impactful, I think several methodological and conceptual aspects could be improved or clarified

1. The paper fixes a single “most discriminative” layer (found using TriviaQA) for all datasets and analyses, which is not justified. While this is stated in limitations, adding an analysis on using layers identified from each dataset will be interesting. Given that computational costs are stated being few minutes on CPU, it seems feasible to repeat Figure 3 experiments using dataset-specific optimal layers (or even Figure 2 with more datasets instead of TriviaQA).
Line 247:
> "which mitigates the risk of discovering an activation direction tied to features merely correlated with model success rather than model's internal correctness prediction.
* 1.1 This is neither theoretically nor empirically justified. Why should there exist a layer that is responsible for model's internal correctness prediction? Are there any theoretical explanations or previous work (other than the empirical results) justifying this?

2. The proposed method is barely better than random chance on GSM8k. While the authors acknowledge this, there is no reasonable explanation. Is this a problem with all reasoning tasks or only mathematical reasoning? Adding additional reasoning tasks to the experiments might help answer this.

3. The "correctness direction" may inadvertently capture memorization rather than self-correctness. All datasets with high performance (TriviaQA and custom datasets) require factual recall that likely overlaps with training data, where as GSM8K (which has a low performance) requires reasoning and likely contains fewer memorized items. The performance drop on GSM8K could therefore be because of absence of memorized associations rather than the absence of correctness awareness. Evaluating and discussing memorization (and potential contamination) effects would strengthen the paper and the claim that such layer exists.

**Questions:**

Re weakness #2 on low performance on GSK8K: One interpretation that I can think about is trivia-style questions like TriviaQA involve retrieving facts, while reasoning tasks like GSM8K require intermediate computation before the model realizes if it can get it correct or not. What do authors think about this?

---

> ### Author Response · Authors · 2025-11-21
> **Rebuttal 1/2**
>
> We thank the reviewer for their interest in our work and attention to detail. We have some responses below to the *comments/weaknesses* and hope that this discussion can increase your confidence in our work.
>
> *The paper fixes a single “most discriminative” layer (found using TriviaQA) for all datasets and analyses, which is not justified. While this is stated in limitations, adding an analysis on using layers identified from each dataset will be interesting. Given that computational costs are stated being few minutes on CPU, it seems feasible to repeat Figure 3 experiments using dataset-specific optimal layers (or even Figure 2 with more datasets instead of TriviaQA).*
>
> and
>
> *Line 247:
> ""which mitigates the risk of discovering an activation direction tied to features merely correlated with model success rather than model's internal correctness prediction.
> 1.1 This is neither theoretically nor empirically justified. Why should there exist a layer that is responsible for model's internal correctness prediction? Are there any theoretical explanations or previous work (other than the empirical results) justifying this?"*
>
> We thank the reviewer for this thoughtful comment and agree that our motivation for the layer-selection procedure was not sufficiently clear. Our goal in Section 4.3.1 was to study cross-dataset generalisation of one fixed “correctness direction” per model. For that purpose, we needed a single consistent layer per model to avoid cherry-picking different layers for different test datasets. We chose TriviaQA for layer selection because it is our largest and most diverse dataset, covering many topics and difficulty levels, which reduces the risk that the optimal layer we identify is specific to a narrow domain.
>
> We do not assume the existence of a unique “correctness layer”. In fact, Figure 9 already shows, for each dataset, the AUROC achieved by the direction at each layer (thus extending Figure 2, as the reviewer suggests). Two empirical patterns follow:
> * Early layers are consistently weak, and performance peaks around the middle-to-late layers.
> * From then on, there is typically a band of adjacent layers that achieve similar AUROC
>
> Thus, correctness-related information appears progressively and is strongest in an intermediate region, and is not localised to a unique “center”. We will emphasise this point more clearly in the main text and refer to Figure 9 earlier. We also report in Appendix B.7 both the “best” layer and the “first good-enough” layer (achieving 90% of the best TriviaQA AUROC), which is consistent with this finding.
>
> Following the reviewer’s suggestion, we will add an additional analysis using dataset-specific optimal layers (i.e., the best layer per dataset, instead of the TriviaQA-selected one) and report the corresponding AUROCs in the appendix. This will take a bit of time, as it will require processing >200 GB of data again so we are queueing up the servers required.
>
> Finally, we do not intend to claim a theoretically special “internal correctness layer”. The results we have so far are empirical and in line with prior findings on truth / knowledge directions (e.g., Burns et al., 2022; Burger et al., 2024; Ferrando et al., 2025; Bao et al., 2025), which also find that relevant signals tend to be strongest in mid-late layers rather than at the input or output.
>
> *The proposed method is barely better than random chance on GSM8k. While the authors acknowledge this, there is no reasonable explanation. Is this a problem with all reasoning tasks or only mathematical reasoning? Adding additional reasoning tasks to the experiments might help answer this.*
>
> and
>
> *Re weakness #2 on low performance on GSK8K: One interpretation that I can think about is trivia-style questions like TriviaQA involve retrieving facts, while reasoning tasks like GSM8K require intermediate computation before the model realizes if it can get it correct or not. What do authors think about this?*
>
> We agree that GSM8K is a crucial part of the story and we see the near-chance performance there as an important finding, not a limitation of our study. In fact, the reviewer’s interpretation matches ours. For trivia-style questions, success is largely governed by whether the relevant fact is encoded and retrievable, this can plausibly be reflected in the activations after reading the question. For GSM8K, we suspect, success depends on a chain of intermediate computations that unfolds during generation. It is therefore natural that a question-only, linear probe struggles to predict correctness on GSM8K, even though it performs well on factual datasets. Most notably, Figure 5 shows that increasing TriviaQA training samples improves performance on other factual datasets, whereas GSM8K performance quickly plateaus, suggesting that the issue is structural rather than data scarcity!
>
> [Cont. below]

---

> > ### Author Response · Authors · 2025-11-21
> > **Rebuttal 2/2**
> >
> > [Cont. above]
> >
> > We also agree that extending the analysis to additional reasoning tasks and probing during generation (e.g., at intermediate chain-of-thought steps) would be very informative. However, these require generating and storing substantially more activations, which is beyond our current compute budget for the rebuttal. We will add a paragraph in the Limitations / Future Work section suggesting correctness probes along the reasoning trajectory (as in Kudo et al., 2024), and adding non-math reasoning benchmarks as important next steps. At a high level though, we believe our current results already provide sufficient evidence for the conclusion: LLMs exhibit a robust, linearly accessible pre-generation correctness signal for factual knowledge, but we do not find such evidence for multi-step mathematical reasoning in the question-only regime.
> >
> > *The "correctness direction" may inadvertently capture memorization rather than self-correctness. All datasets with high performance (TriviaQA and custom datasets) require factual recall that likely overlaps with training data, where as GSM8K (which has a low performance) requires reasoning and likely contains fewer memorized items. The performance drop on GSM8K could therefore be because of absence of memorized associations rather than the absence of correctness awareness. Evaluating and discussing memorization (and potential contamination) effects would strengthen the paper and the claim that such layer exists.*
> >
> > We appreciate the concern about memorisation, and indeed believe that correctly pinning down construct validity is really important for getting probes right. First, we would like to clarify what our probe is doing. The probe is trained to separate internal states that precede correct vs incorrect answers for the same model, not to recognise particular questions. Even if the underlying factual content was originally learned via memorisation, the direction is capturing when the model is about to use that knowledge successfully (including near-misses and “I don’t know” refusals), which we view as closer to an internal confidence/correctness signal than to direct benchmark memorisation. Moreover, several of our datasets (Cities, Notable People, Medals, Math Ops) are synthetically built from public tables with our own templates, so exact benchmark contamination is excluded, and yet the direction trained on TriviaQA transfers well to them. GSM8K, by contrast, is structurally different: success hinges on multi-step reasoning rather than simple recall, and it is likely not systematically “less seen” in pretraining than our factual sources. We agree that we do not perform a dedicated contamination analysis, and believe that future work should focus on disentangling memorisation from correctness awareness (this coud be done through a variety of ways, such as fictional entities introduced only in the prompt, systematic perturbations of entities/years, and membership-inference comparisons). We will include in the limitations section of our work, and emphasise that our claim is about a robust pre-generation factual correctness signal in mid-layers, which is not a memorisation-free notion of correctness.

---

> > > ### Author Response · Authors · 2025-11-27
> > > **Rebuttal 3/3**
> > >
> > > We have now run the dataset-specific layer search for most datasets (within constraints of data storage) and all models; see new Table 8 below (will be uploaded in App. B.7). We find that AUROCs with dataset-optimal layers are only a few points higher than with the single TriviaQA-selected layer, and the cross-dataset pattern is unchanged.
> > >
> > > We believe this provides evidence that using one model-optimal layer is already highly effective and not hiding a radically different dataset-specific optimum.
> > >
> > > Table 8. AUROC values on the model optimal layer from Section \ref{sec: find_layer} and on the layer optimized per dataset for all models and datasets.
> > > | Model | Metric | TriviaQA (Model) | TriviaQA (Dataset) | Cities (Model) | Cities (Dataset) | Math ops. (Model) | Math ops. (Dataset) | Medals (Model) | Medals (Dataset) | GSM8K (Model) | GSM8K (Dataset) |
> > > |:---:|:---:|:---:|:---:|:---:|:---:|:---:|:---:|:---:|:---:|:---:|:---:|
> > > | Llama 3.1 8B | Layer | 14 | 16 | 14 | 30 | 14 | 12 | 14 | 28 | 14 | 12 |
> > > | | AUROC | 0.802 | 0.803 | 0.732 | 0.783 | 0.860 | 0.901 | 0.677 | 0.712 | 0.537 | 0.548 |
> > > | Llama 3.3 70B | Layer | 76 | 76 | 76 | 72 | 76 | 32 | 76 | 40 | 76 | 32 |
> > > | | AUROC | 0.821 | 0.821 | 0.881 | 0.890 | 0.835 | 0.979 | 0.772 | 0.794 | 0.499 | 0.625 |
> > > | Qwen 2.5 7B | Layer | 22 | 22 | 22 | 20 | 22 | 20 | 22 | 20 | 22 | 20 |
> > > | | AUROC | 0.760 | 0.760 | 0.841 | 0.854 | 0.833 | 0.875 | 0.587 | 0.588 | 0.602 | 0.678 |
> > > | Deepseek R1 32B | Layer | 44 | 44 | 44 | 48 | 44 | 52 | 44 | 48 | 44 | 12 |
> > > | | AUROC | 0.734 | 0.734 | 0.880 | 0.890 | 0.848 | 0.963 | 0.640 | 0.665 | 0.554 | 0.642 |
> > > | Mistral 7B | Layer | 16 | 16 | 16 | 30 | 16 | 22 | 16 | 20 | 16 | 18 |
> > > | | AUROC | 0.798 | 0.798 | 0.879 | 0.914 | 0.801 | 0.915 | 0.647 | 0.699 | 0.579 | 0.592 |
> > > | Ministral 8B | Layer | 18 | 18 | 18 | 16 | 18 | 16 | 18 | 16 | 18 | 2 |
> > > | | AUROC | 0.738 | 0.738 | 0.840 | 0.846 | 0.843 | 0.859 | 0.666 | 0.669 | 0.574 | 0.637 |
> > >
> > > We will also create a new summary figure with the full out-of-distribution setup, which should hopefully make the full story of the paper easier to absorb!

---

### Official Review · Reviewer_BTu1 · 2025-10-31

**Soundness:** 3
**Presentation:** 3
**Contribution:** 3
**Rating:** 4
**Confidence:** 3

**Summary:**

This paper investigates whether large language models (LLMs) already "know" the correctness of their answers within their internal activations before responding. The study finds that by analyzing the model's activation state after reading a question using a simple linear probe, it is possible to predict the accuracy of its answer in advance for knowledge-based questions. Furthermore, this method demonstrates superior generalization performance compared to baselines.

**Strengths:**

1. Novel Approach: The research focuses on predicting answer correctness solely from internal model activations before any tokens are generated. This differs from most studies that detect hallucinations or errors after generation, offering a new perspective on understanding the model's internal cognitive processes.
2. Simple and Effective Method: The study achieves results using only a simple linear probe (calculating the mean difference between correct/incorrect activations). This demonstrates that the "correctness" signal is linearly separable within the model, representing a clear and fundamental representation that doesn't necessarily require a complex classifier.
3. Strong Generalization: A probe trained on a general knowledge base (TriviaQA) can effectively generalize to other unseen knowledge domains. Its performance significantly surpasses black-box evaluators and model self-consistency methods, proving that internal activations contain unique and valuable signals.

**Weaknesses:**

1. Narrow Scope of Application: The probe's core capability is limited to knowledge-retrieval questions. The paper also explicitly states the method is almost ineffective for mathematical reasoning tasks (GSM8K) that require multi-step computation, indicating this "pre-cognition" ability is highly limited.
2. Inconsistent Signal Across Layers: The appendix (e.g., Figure 9) shows that the signal peak for different task types appears in different model layers. This suggests the model may lack a unified "self-awareness" center, making the practical deployment of a single, simple probe unstable.
3. Oversimplified Definition of Correctness: The study treats correctness as a binary label and operates in a deterministic (zero-temperature) setting. This ignores the stochasticity of answers and the ambiguity of correctness in real-world scenarios.

**Questions:**

1. Why does it fail on reasoning? This is the most critical question. Is it because the correctness signal for reasoning only emerges during the generation process, or is this type of signal inherently non-linear? This points to an important direction for future research.
2. Practical value of "white-box" access? This method requires access to internal model activations and necessitates training a probe for a specific model and layer. In practical deployment, what are its main advantages in terms of computational overhead and maintenance costs compared to post-generation detection methods? Furthermore, since the main experiment only proves its generalization is significantly better than the baseline, but poorer performance on the corresponding datasets, does this also imply the method might just be a mediocre choice rather than the optimal solution?

---

> ### Author Response · Authors · 2025-11-21
> **Rebuttal 1/3**
>
> We thank the reviewer warmly for their attention to detail. Below we respond to each *comment/weakness*, and hope that you can reconsider your score in light of our discussion.
>
> *Narrow Scope of Application: The probe's core capability is limited to knowledge-retrieval questions. The paper also explicitly states the method is almost ineffective for mathematical reasoning tasks (GSM8K) that require multi-step computation, indicating this "pre-cognition" ability is highly limited.*
>
> We agree that our method is effective primarily on knowledge-style tasks, and we see this not as a defect of the setup but as an important scientific result. The aim of the paper is not to propose a plug-and-play deployment tool (as indeed this may already be the objective of some startups or commercial setups), but to characterise what types of internal correctness signals exist in current LLMs even under a controlled “question-only” setting. Concretely, we show that a simple linear direction trained on TriviaQA generalises well across several factual knowledge domains, but does not transfer to GSM8K. This gives some evidence that models store a relatively accessible “knowledge correctness” signal before generation, while the success of multi-step mathematical reasoning depends on computations that are not yet encoded at question-only time, or are not linearly separable. We will clarify this framing in the Introduction and Discussion so that readers do not interpret the limited GSM8K performance as a failure of the method, but rather as a boundary of the phenomenon we are studying.
>
> *Inconsistent Signal Across Layers: The appendix (e.g., Figure 9) shows that the signal peak for different task types appears in different model layers. This suggests the model may lack a unified "self-awareness" center, making the practical deployment of a single, simple probe unstable.*
>
> We did not intend to claim that there is a single “self-awareness center” layer. In fact, many interpretability works (such as the ones we cite in our related works) find that useful high-level features emerge gradually and are strongest in middle to late layers rather than being localised to a single layer. Our results show a fairly consistent pattern across models: early layers perform poorly, the correctness signal rises around the middle, and a range of nearby layers achieve similar AUROC (Figure 9). In the main text we explicitly select the best layer on TriviaQA, **but importantly in Appendix B.7 we also report the “first good enough” layer that reaches 90% of the best TriviaQA performance, and this lies in a fairly broad band.** This suggests that the signal is distributed across several adjacent layers.
>
> Moreover, the fact that the peak performance for some of the ""specific"" datasets (e.g., Math Operations) appears in different layer from the peak for TriviaQA is likely due to the linear probe for the specific dataset learning dataset-specific features that are connected to performance on that dataset but may not transfer to other datasets, which makes them poorly generalisable (as can be seen in Figure 6).
>
> For practical use, one could aggregate projections from a small window of layers or simply choose the first “good enough” layer instead of the exact optimum.
>
> *Oversimplified Definition of Correctness: The study treats correctness as a binary label and operates in a deterministic (zero-temperature) setting. This ignores the stochasticity of answers and the ambiguity of correctness in real-world scenarios.*
>
> This is right! We agree that the fully-fledge construct of correctness can be stochastic and graded in realistic deployments, and we do not claim that our binary, temperature-0 setup fully captures this complexity. We intentionally chose a highly controlled setting in order to isolate the existence and structure of a latent correctness signal, so as to avoid conflating epistemic uncertainty with some more unwieldy artifacts such as sampling noise or ambiguous labels. Zero temperature and binary labels remove aleatoric randomness from the target, which lets us ask (what we feel is) a fairly clean research question: “Given this deterministic policy, are activations before generation linearly predictive of whether the model’s chosen answer will match the gold label?” We see this as a necessary first step before tackling more complex regimes that involve multiple samples per question, soft correctness scores, or non-binary grading, which some works may have jumped to already, and discuss this as a limitation in Section 6. In the revision, we will strengthen that discussion and explicitly state that extending the analysis to stochastic decoding and graded correctness is an important direction for future work, not addressed by the current paper.

---

> > ### Author Response · Authors · 2025-11-21
> > **Rebuttal 2/3**
> >
> > *Why does it fail on reasoning? This is the most critical question. Is it because the correctness signal for reasoning only emerges during the generation process, or is this type of signal inherently non-linear? This points to an important direction for future research.*
> >
> > We fully agree that understanding the GSM8K failure is central and we thank the reviewer for articulating the key hypotheses. At the moment, our current experiments cannot definitively distinguish between “signal only emerges during generation” and “signal is present but non-linear,” so we are careful not to over-interpret the result. Our findings suggest two plausible explanations:
> >
> > **Temporal explanation.** For multi-step reasoning, whether the model will succeed depends on the internal chain of computations that unfolds during generation. At question-only time, the representation may encode problem type and surface difficulty, but not the detailed trajectory of reasoning, so a correctness signal may only become linearly separable later in the chain-of-thought.
> >
> > **Geometric / non-linear explanation.** It is also possible that for reasoning tasks the relevant correctness features exist at question-only time but are not well approximated by a single linear direction. This is consistent with our sample-size study (Figure 5), where increasing training data for GSM8K does not close the performance gap, suggesting that limited data is not the main issue.
> >
> > A natural extension here is to repeat our analysis along the full generation trajectory, or to compare linear probes with higher-capacity non-linear probes on reasoning tasks. We see our GSM8K result as pointing directly to this future work, and add this to our limitations section as future work.
> >
> > *Practical value of "white-box" access? This method requires access to internal model activations and necessitates training a probe for a specific model and layer. In practical deployment, what are its main advantages in terms of computational overhead and maintenance costs compared to post-generation detection methods? Furthermore, since the main experiment only proves its generalization is significantly better than the baseline, but poorer performance on the corresponding datasets, does this also imply the method might just be a mediocre choice rather than the optimal solution?*
> >
> > Our primary goal in this paper is to gain scientific insight into what current LLMs encode internally, rather than to propose a single best practical detector. That said, white-box probes of the kind we study do have concrete advantages in settings where internals are available, such as open models or on-prem deployments. Once activations are cached, our probe reduces to a single linear projection per example. Training the probe means learning one d-dimensional vector on cached activations, which takes minutes on CPU, and inference is much cheaper than any sampling-based or post-generation method that requires multiple answers. This makes it a natural choice for a pre-generation or early-warning filter under tight token budgets, which can be followed by more expensive post-generation checks on questions flagged as uncertain.
> >
> > We have some new results which provide some evidence for this, actually! In Table 7 (Appendix B.3), we report accuracies for a thresholded classifier based on the correctness direction, with a single scalar threshold chosen on the training dataset and then transferred to other datasets. This shows that the probe can be used as a simple, concrete decision mechanism, and that it retains the robustness patterns suggested by AUROC. While some black-box assessors achieve slightly higher in-distribution AUROC on TriviaQA, our results show that they overfit and degrade substantially under domain shift, whereas the correctness direction generalises much better to new factual datasets. For deployments where safety or distribution shift is a significant concern, this can matter more or less than marginal in-distribution gains, and our probe may not be the optimal practical solution in isolation. Rather, we see it as a complementary white-box tool that reveals a correctness signal in model internals and can form one component of a broader ensemble that combines white-box and black-box uncertainty estimators.

---

> > > ### Author Response · Authors · 2025-11-21
> > > **Rebuttal 3/3**
> > >
> > > Below is the Table 7 mentioned above.
> > >
> > > Table 7. Accuracy of a classifier based on the correctness direction at the optimal layer. For each test dataset, we display results for the direction trained on TriviaQA and for the direction trained on the dataset itself. The threshold for the classifier is the optimal one for the training data.
> > > | Model            | TriviaQA (TQA train) | TriviaQA (self train) | N. people (TQA train) | N. people (self train) | Cities (TQA train) | Cities (self train) | Math ops. (TQA train) | Math ops. (self train) | Medals (TQA train) | Medals (self train) | GSM8K (TQA train) | GSM8K (self train) |
> > > |------------------|----------------------|------------------------|------------------------|-------------------------|--------------------|----------------------|------------------------|-------------------------|--------------------|----------------------|--------------------|----------------------|
> > > | Llama 3.1 8B     | 0.728                | 0.728                  | 0.522                  | 0.688                   | 0.551              | 0.714                | 0.755                  | 0.874                   | 0.612              | 0.682                | 0.500              | 0.670                |
> > > | Llama 3.3 70B    | 0.750                | 0.750                  | 0.621                  | 0.651                   | 0.797              | 0.808                | 0.520                  | 0.943                   | 0.553              | 0.782                | 0.500              | 0.671                |
> > > | Qwen 2.5 7B      | 0.688                | 0.688                  | 0.639                  | 0.764                   | 0.515              | 0.762                | 0.792                  | 0.938                   | 0.512              | 0.640                | 0.501              | 0.654                |
> > > | Deepseek R1 32B  | 0.671                | 0.671                  | 0.588                  | 0.793                   | 0.612              | 0.802                | 0.848                  | 0.961                   | 0.519              | 0.636                | 0.500              | 0.660                |
> > > | Mistral 7B       | 0.721                | 0.721                  | 0.523                  | 0.724                   | 0.771              | 0.811                | 0.724                  | 0.858                   | 0.545              | 0.661                | 0.503              | 0.659                |
> > > | Ministral 8B     | 0.670                | 0.670                  | 0.514                  | 0.647                   | 0.500              | 0.774                | 0.764                  | 0.830                   | 0.532              | 0.656                | 0.502              | 0.671                |

---

### Official Review · Reviewer_DabN · 2025-10-31

**Soundness:** 2
**Presentation:** 3
**Contribution:** 1
**Rating:** 2
**Confidence:** 4

**Summary:**

The paper studies LLM probing (linear probing) on the first generation token, i.e., before the answer is generated. The experiments are done for various LLMs. The authors study generalization and obtain decent performance for TriviaQA and some simplistic synthetic factual QA datasets, but esp. transfer seems to fail for the mathematical reasoning data. The authors also show some qualitative examples (e.g., questions where the LLM answers IDK in the extreme of the correctness direction).

**Strengths:**

- LLM probing is an interesting topic
- The paper is well-written
- The missing transfer to reasoning data is an interesting observation, but no more analysis is provided. It could also be more generally that it's data different from factual QA.

**Weaknesses:**

- Given that verbalized confidence is known to be uncalibrated it's a very weak baseline.
- In terms of factual data, where the answers are short, the question-only setting does not seem to have considerable efficiency advantage, compared to the one with answer generation.
- The paper completely misses critically-related work. There are other papers considering the same kind of probing: [1-5]. Those papers show some very similar results.
- I consider the provided, very simplistic experiments not very novel/innovative, esp. in the light of the related work. Overall, the insights are in my opinion not enough to justify presentation at ICLR.

----------

[1] Snyder et al. "On early detection of hallucinations in factual question answering." Proceedings of the 30th ACM SIGKDD Conference on Knowledge Discovery and Data Mining. 2024.

[2] Gottesman et al. "Estimating knowledge in large language models without generating a single token." arXiv preprint arXiv:2406.12673 (2024).

[3] Wang et al. "Hidden question representations tell non-factuality within and across large language models." arXiv e-prints (2024): arXiv-2406

[4] Ji et al. "Llm internal states reveal hallucination risk faced with a query." arXiv preprint arXiv:2407.03282 (2024)

[5] Slobodkin et al. The Curious Case of Hallucinatory (Un)answerability: Finding Truths in the Hidden States of Over-Confident Large Language Models, EMNLP'23

**Questions:**

-----

---

> ### Author Response · Authors · 2025-11-21
> **Rebuttal 1/2**
>
> We thank the reviewer warmly for their attention to detail and prior work. Below we respond to each *weakness*, and hope that you can reconsider your score in light of our discussion.
>
> *Given that verbalized confidence is known to be uncalibrated it's a very weak baseline.*
>
> We agree that verbalised confidence alone is not a strong baseline! This is precisely why we also include assessor baselines (logistic regression and XGBoost) trained on model-independent question embeddings (Sec. 4.1.3, App. B.3), which are stronger and widely used in the predictability literature (e.g., Hernández-Orallo et al., 2022; Pacchiardi et al., 2024; 2025). On factual datasets, our correctness direction consistently outperforms both verbalised confidence and the best assessor out-of-distribution (Table 3), with gains of 5–39 AUROC points (depending on dataset and model), while assessors sometimes perform better in-distribution but overfit (App. B.3). Furthermore, our goal is not to claim a new state-of-the-art calibration method, but to ask whether a one-dimensional direction in the residual stream can meaningfully predict correctness compared to standard external baselines.
>
> *In terms of factual data, where the answers are short, the question-only setting does not seem to have considerable efficiency advantage, compared to the one with answer generation.*
>
> We see the question-only setting primarily as a scientific probe rather than as a deployment scheme (which would likely employ some mixture of white- and black- box methods). As we state in the introduction, our main aim is to test whether residual stream activations immediately after reading the question contain a latent correctness signal, not to minimise inference cost. That said, once activations are available (e.g., because the model is already being called for another purpose), our probe is extremely cheap: learning the d-dimensional direction is a one-shot computation on cached activations (<3 minutes on CPU for 10k samples), and applying it is a single linear projection per question (Sec. 4.1). By contrast, several existing uncertainty methods require multiple generations or additional prompts.
>
> *The paper completely misses critically-related work. There are other papers considering the same kind of probing: [1-5]. Those papers show some very similar results.*
>
> We thank the reviewer for pointing out [1–5]; these works are indeed closely related and have integrated them into Further Related Work and Appendix A. Taken together with our results, we think they support a consistent picture and show that internal states at question time already contain information about future factuality or hallucination risk. Our contribution in this cluster of works is to show that this signal can be captured in a particularly constrained and interpretable form, and to characterise where it generalises and where it fails.
>
> Concretely, our work differs from [1–5] in three main ways which we believe are important:
>
> **Stricter and simpler probe.** Several of [1–5] use richer features of the generation process (e.g. early generated tokens, multi-step trajectories, or question+context encodings) and more expressive classifiers. We deliberately work in a stricter setting: a single one-dimensional difference-of-means direction in the residual stream of one layer, computed from question-only activations before any token is generated. Showing that this minimal probe already predicts correctness across datasets and models strengthens the case that correctness/confidence is a linearly accessible internal quantity, not just something recoverable with complex readouts. We build this case showing that it even has advantages against more advanced classifier baselines, which we hope makes clear that our choice of a minimal setting was deliberately considered.
>
> **Systematic cross-dataset and cross-model analysis of self-correctness.** Several cited works target hallucination or answerability in reading comprehension with context passages, or knowledge at the entity level (e.g. “all questions about Napoleon”). We instead study per-question self-correctness for free-form factual QA across six open-source models from three families (7B–70B), and we do so with a fixed layer per model chosen on TriviaQA. We then systematically map how a single TriviaQA-trained direction transfers to various factual domains (Table 3, Fig. 3, App. B.5–B.6), including failures on GSM8K. We think this experimental design gives a clearer picture of how broad the internal correctness signal is and when it breaks, which is our focus in this work.
>
> [Cont. below]

---

> > ### Author Response · Authors · 2025-11-21
> > **Rebuttal 2/2**
> >
> > [Cont. from above]
> >
> > **Characterising failure modes and linking to abstention behaviour.** To our knowledge, prior work has not jointly: (i) shown that the question-only correctness signal fails structurally on multi-step mathematical reasoning even with more data (Fig. 5), and (ii) connected the same direction to spontaneous “I don’t know” answers and near-miss errors (Fig. 4, Table 4). This matters both scientifically and for safety: it suggests current models internally track their likelihood of factual success, but that this tracking does not straightforwardly extend to chain-of-thought style reasoning tasks.
> >
> > We will revise the paper to explicitly position our results as consolidating and sharpening this emerging line of work: by using a very simple, reproducible linear probe in a strict question-only setting, across multiple open models and datasets, we provide clean evidence for a mid-layer “correctness/confidence” axis, and we delineate its limits on reasoning tasks. Below is the additional section we would include in our related work:
> >
> > >Snyder et al. and Ji et al. focus on hallucination risk using rich features of the generation process, often including the first generated token, whereas we intentionally work in a stricter setting where a single one-dimensional direction in the residual stream is used to predict success before any token is generated. Slobodkin et al. focus on (un)answerability in reading comprehension with an explicit context passage, while we target free-form factual question answering and label each example by whether the specific answer produced under greedy decoding is correct. Gottesman and Geva focus on knowledge at the level of entities, such as “all questions about Napoleon”, while we study per-question self-correctness and show that a single probe trained on TriviaQA transfers across diverse knowledge datasets and across three open-source model families and sizes. Wang et al., we believe, are closest in spirit. Compared to our work, they train significantly more expressive probes for non-factuality and focus on cross-model domain adaptation, while we restrict ourselves to a simple difference-of-means probe, provide a systematic layer-wise analysis that identifies where the signal is strongest, and do significant work to characterise failure modes in reasoning, and link the same direction to spontaneous “I don’t know” responses.
> >
> > *I consider the provided, very simplistic experiments not very novel/innovative, esp. in the light of the related work. Overall, the insights are in my opinion not enough to justify presentation at ICLR.*
> >
> > We agree that our methodology is deliberately simple. This is by design! Our contribution is to show that a minimal linear probe on question-only residual stream activations already yields a robust, mid-layer correctness / confidence signal that (i) generalises across multiple factual domains and six open-source models (7B–70B), (ii) fails systematically on GSM8K in a way that persists even with more data (Fig. 5), and (iii) aligns with abstention behaviour (“I don’t know”) and near-miss errors (Secs. 4.4–4.6, Table 4). We complement earlier work by providing: a careful layer-wise analysis of where this signal emerges (Fig. 2, Fig. 9), train–test heatmaps over all dataset combinations that distinguish dataset-specific cues from genuinely general signals (Fig. 3, App. B.5–B.6), and a systematic comparison with both verbalised confidence and strong assessor baselines in- and out-of-distribution (Table 3, App. B.3). We see our results as still very useful: together with [1–5] and the SAE-based work we cite, they provide evidence across models and against several baselines of a converging story/hope(?) in mechanistic interpretability. Namely, that large models encode an internal, linearly accessible notion of correctness/confidence, while also highlighting clear limitations on multi-step reasoning.

---

> ### Comment · Reviewer_DabN · 2025-11-23
>
> Thank you for the response. I'll provide more justification for my rating below (but don't want to start a discussion on those).
>
> **Related Works**
> I consider the omission of all these related in the original submission as questionable scientifically. It is normal to miss some related work in the flood of AI literature we're facing today. But barely referencing any is strange. I'd also expect a careful delimitation; e.g., instead of "Several of [1–5]", mention the exact ones, and mention also which ones do it differently.
>
> **Simple Experiments**
> I appreciate simple and elegant models, but in my review I was referring to the evaluation methodology. Since the topic has been studied before, I would have expected more detailed analysis (e.g., in comparison to the models in the works mentioned). The insights about the "I don't know" etc. are interesting, but in my opinion not enough.

---

> > ### Author Response · Authors · 2025-11-25
> > **Rebuttal**
> >
> > We thank the reviewer for their continued engagement with our work.
> >
> > > **Related Works** I consider the omission of all these related in the original submission as questionable scientifically. It is normal to miss some related work in the flood of AI literature we're facing today. But barely referencing any is strange.
> >
> > We respectfully disagree with the characterization that we "barely referenced any" related work. Our original submission included approximately 12 works employing model internals to predict various prompt features, discussed in the related works section and detailed extensively in Appendix A. While we acknowledge we were unaware of the specific five works the reviewer mentioned (which is indeed common given the current pace of AI publications) we believe this does not constitute "questionable scientifically" conduct. We have now integrated all five works into our revision.
> >
> > Moreover, while related and worth adding (as we've done in our revision), none of the works the reviewer mentioned exactly employs "the same kind of probing", as the reviewer claims (see below).
> >
> > > I'd also expect a careful delimitation; e.g., instead of "Several of [1–5]", mention the exact ones, and mention also which ones do it differently.
> >
> > We report here, for the reviewer's convenience, the text we had added to the related works appendix at the time of our first response:
> >
> > > Several contemporaneous works also use LLM internals to anticipate hallucination or non-factuality
> > at question time (Snyder et al., 2024; Gottesman & Geva, 2024; Wang et al., 2024; Ji et al., 2024;
> > Slobodkin et al., 2023). Snyder et al. (2024) and Ji et al. (2024) focus on hallucination risk and
> > often use richer information about the generation process, such as the first generated token or short
> > trajectories. Slobodkin et al. (2023) study (un)answerability in reading comprehension with an
> > explicit context passage. Gottesman & Geva (2024) shift the focus to the level of entities, and
> > estimate whether a model “knows” facts about specific entities such as historical figures. Wang
> > et al. (2024) are closest in spirit to our work, and train more expressive classifiers over internal
> > states to predict non-factuality and to transfer signals across models. In contrast, we deliberately
> > adopt a stricter and simpler setting. We work with a single one-dimensional difference-of-means
> > direction in the residual stream of one layer, trained on question-only activations before any token is
> > generated, and we study per-question self-correctness for free-form factual QA across six open-source
> > models from three families. This minimal probe already predicts correctness across datasets and
> > models, which strengthens the claim that a linearly accessible correctness or confidence signal exists
> > in mid–late layers.
> >
> > Our phrase "Several of [1-5]" in the initial rebuttal was informal shorthand; the full text we added (quoted above) specifies which works do what.
> >
> > > Simple experiments [...]
> >
> > We appreciate the clarification that the concern is about evaluation methodology rather than model complexity. We would respectfully highlight that our experimental work is more comprehensive than it may initially appear. Our analysis spans 6 models x 6 datasets x up to 80 layers, with 5-fold cross-validation throughout. The train-test heatmaps alone (Figure 3, Appendix B.5) cover 216 experimental conditions to try our very hardest to systematically distinguish dataset-specific cues from generalizable signals. Beyond measuring where the probe succeeds, we carefully characterize where it fails. Figure 5 demonstrates that GSM8K performance plateaus regardless of training data size (from 20 to 48,540 samples, with 10 repetitions per condition to reduce variance), ruling out data scarcity and pointing to a structural limitation. We further connect the correctness direction to spontaneous abstention behavior (Figure 4, Section 4.5), showing that ""I don't know"" responses cluster at the negative extreme.
> >
> > Regarding direct comparison to the methods in [1-5]: because these works use richer features (generation trajectories, context passages) or more expressive classifiers, they are effectively solving different problems, making AUROC comparisons less informative. Our contribution is precisely to show that a minimal probe under strict question-only constraints already recovers a robust, generalizable signal which is structurally limited in some senses. This strengthens the scientific claim that correctness/confidence is linearly accessible in model internals, and we see our findings as complementary to [1-5], collectively building converging evidence for the same underlying phenomenon.

---

> ### Comment · Reviewer_DabN · 2025-11-27
>
> Thank you. I confirm that I've read this and will discuss with the other reviewers.

---

### Author Response · Authors · 2025-12-03
**Additional framing, experiments, and results**

To the Area Chair,

As suggested by the program chairs following the reversion of the discussion phase, we submit this summary to ensure you have the complete picture of the generative (enjoyable!) discussion that took place. We engaged with all four reviewers, adding significant new experiments and substantially revising our manuscript in response to their feedback. We appreciated the feedback that our work offers "*a new perspective on understanding the model's internal cognitive processes*," that the results are "*consistent across all models*" across "*varying sizes and architectures*," and most of all that "*the paper includes sufficient detail to understand the complete experimental and methodological design*" while being "*well written*".

We now believe the paper is considerably stronger than the initial submission. Below we summarize the big updates:

1. Major Reframing: Testing the Linear Representation Hypothesis

Reviewer vgab provided feedback that our original framing overemphasized "beating baselines" rather than the scientific question. We thought this was a well thought-out comment, and have fully accepted this critique. We have restructured the Introduction, Methodology, and Conclusion accordingly.

**New framing.** The paper now explicitly says how our aim is to test whether a linearly separable "correctness direction" exists in the residual stream before generation begins. The core scientific contribution is demonstrating that this signal:
* Exists and is linearly accessible in mid-to-late layers
* Generalizes across diverse factual knowledge domains
* Fails on mathematical reasoning (GSM8K), and in a way that suggests it is structural rather than due to a failure in elicitation or affordances

**Clarification on baselines.** We believe there was a misunderstanding regarding our baseline comparisons that is worth stating explicitly. Our Assessor baselines use external OpenAI embeddings with logistic regression and XGBoost classifiers. Our probe uses internal model activations with a simple linear direction. These operate on different feature spaces. The finding that a minimal linear probe on internal activations outperforms more expressive non-linear classifiers on external embeddings is the evidence that the residual stream contains unique, linearly-accessible self-competence information not captured by model-independent question features.

2. New Computationally Intensive Experiments

We executed substantial new compute jobs during the rebuttal period to directly address empirical questions raised by reviewers.

**Dataset-specific layer selection (New Table 8).** Reviewer xEzz questioned whether selecting layers based solely on TriviaQA was methodologically sound. We re-processed >200GB of activation data to identify optimal layers for each dataset individually across all six models (in particular, for a given test dataset, we identified the optimal layer where the direction trained on TriviaQA had the largest performance). We found that performance gains from this were negligible (on average, around 1-3 AUROC points). This provides empirical validation for our methodology: the correctness signal consistently saturates in the same band of middle-to-late layers across diverse tasks, supporting the existence of a general phenomenon rather than dataset-specific artifacts. We also brought this up in conjunction with existing experiments in the appendix showing results for the first "good enough" layers, which would be relevant for a production setup using such a classifier.

**Practical accuracy with fixed thresholds (New Table 7).** Reviewers vgab and BTu1 requested practical metrics beyond AUROC. We computed accuracy scores using fixed decision boundaries chosen only on training data. We found our probe achieves high practical accuracy (~80-96% on synthetic factual tasks, ~67-75% on TriviaQA), demonstrating it functions as a reasonable deployable classifier.

**Ongoing work.** We are currently running, but have not yet finished a few experiments. Namely, we are working on (1) adding synthetic noise, (2) training assessors on the other datasets, and (3) producing the data needed for a comparison boxplot/heatmap.

[Continued below.]

---

> ### Author Response · Authors · 2025-12-03
> **Additional framing, experiments, and results (part 2)**
>
> 3. Comprehensive Experimental Scope (Reviewer DabN)
>
> Reviewer DabN characterized our experiments as simplistic. We respectfully but firmly push back on this characterization and ask the AC to consider the full scope of our work on the following grounds.
>
> **Scale.** We tested 6 models × 6 datasets × up to 80 layers × 5-fold cross-validation throughout. The train-test heatmaps alone (Figure 3, Appendix B.5-B.6) cover 216 experimental conditions to systematically distinguish dataset-specific cues from generalizable signals. Making sense of these experimental results was not straightforward, but important to verify the breadth and robustness of the linear separability of this signal across model scales, families, and settings.
>
> **Simplicity is not a reason to reject.** In mechanistic interpretability, we believe that demonstrating a phenomenon can be captured by a simple linear direction is a stronger scientific result than showing it requires a complex non-linear classifier. Our experimental design deliberately uses a minimal probe (difference-of-means direction) in a strict question-only setting to isolate the existence of the signal. This, we believe, is a feature, not a limitation.
>
> 4. Related Work
>
> **On related work.** Reviewer DabN stated that the omission of five specific works was questionable scientifically. We respectfully but firmly push back on this characterization. We find the suggestion that we "barely referenced any" related work to be inaccurate. In fact, our original submission contained a comprehensive appendix section (Appendix A) with three subsections positioning our work against around 30 citations, even prior to integrating their suggestions. Nonetheless, we considered this an opportunity to differentiate our contributions better, and have integrated all five works carefully. We compare each one suggested individually against our methodology:
> * Snyder et al. and Ji et al. use richer generation features (early tokens, trajectories)
> * Slobodkin et al. focus on answerability in reading comprehension with context passages
> * Gottesman & Geva operate at the entity level ("does the model know about Napoleon?")
> * Wang et al. use more expressive classifiers and focus on cross-model transfer
> and find that our work is distinguished by: (i) a strict question-only, single-layer, linear-direction setting to make the strongest possible claim we can about linear accessibility; (ii) systematic cross-dataset and cross-model analysis (6 models, 7B-70B, three families); (iii) explicit characterization of failure modes on reasoning; and (iv) linking the same direction to spontaneous abstention behavior.
>
> We hope the AC will consider these substantial improvements, the depth and earnestness of our engagement with all reviewers, and the scientific contribution of cleanly demonstrating a linearly accessible correctness signal in LLM internals.
>
> Best regards, The Authors

---

### Meta-Review · Area_Chair_RAzt · 2025-12-30

**Summary:**

The paper investigates whether LLMs possess an internal pre-cognition of their own success by probing residual stream activations immediately after a question is read but before generation begins. Using a simple difference-of-means linear probe, the authors find a correctness direction that generalizes across factual datasets and model families (7B–70B).

The primary concern in the reviews centered on scientific vs. engineering contribution. Reviewers questioned the novelty given existing work on hallucination detection and expressed concerns regarding the lack of performance on reasoning tasks (GSM8K) and the perceived simplicity of the experiments.

**Reviewer Concerns:**

Addressed concerns:
- Authors provided new experiments (Table 8) showing that choosing a model-optimal layer (TriviaQA) vs. dataset-optimal layers results in negligible performance differences, proving the signal is stable across a mid-layer band
- The authors added Table 7, providing hard accuracy numbers and fixed decision boundaries, moving beyond purely threshold-free AUROC metrics to show the probe's potential as a deployable filter
- The authors committed to releasing the synthetic datasets and cached activations, mitigating concerns about the private nature of the custom factual datasets

Outstanding concerns:
- The failure on GSM8K remains an empirical observation. While the authors provided a compelling temporal explanation (that reasoning success depends on the computation path during generation), this remains a hypothesis for future work rather than a resolved mechanism
- Based on author's claim for their novelty and contribution, the paper title and abstract need to modified to focus on knowledge-related dataset rather than all question-related tasks, as datasets which need further reasoning and thinking cannot fit into this work's scope and conclusion
- As mentioned in the reviewer DabN and vgab, previous work has already done similar research and found that LLM’s hidden states/residual streams contain latent confidence estimates. Authors need to reframe their research quesstions to investigate the deeper reasons why the trained linear probe can achieve higher performance only on knowlege-dense questions rather than reasoning-based questions, which can bring more insighrs and contribution to the interpretability community.

**Reviewer Scores:**

Based on the outstanding concerns raised by all reviewers and core limitation of this work, reviewer scores might not changed.

---

### Decision · Program_Chairs · 2026-01-26

Reject